# Physics-based evolution of transmembrane helices reveals mechanisms of cholesterol attraction

Jeroen Methorst [1,2], Nino Verwei [1], Christian Hoffmann [3], Paweł Chodnicki [4,5], Roberto Sansevrino[3], Partha Pyne[6], Han Wang [3], Niek van Hilten [1,7,8], Dennis Aschmann[1], Alexander Kros [1], Loren Andreas[6], Jacek Czub [4], Dragomir Milovanovic [3,9] & Herre Jelger Risselada [1,2] ✉

The existence of linear cholesterol-recognition motifs in transmembrane domains has long been debated. Evolutionary molecular dynamics (Evo-MD) simulations–genetic algorithms guided by (coarse-grained) molecular force-fields–reveal that thermodynamic optimal cholesterol attraction in isolated alpha-helical transmembrane domains occurs when multiple consecutive lysine/arginine residues flank a short hydrophobic segment. These findings are supported by atomistic simulations and solid-state NMR experiments. Our analyses illustrate that linear motifs in transmembrane domains exhibit weak binding affinity for cholesterol, characterized by sub-microsecond residence times, challenging the predictive value of linear CRAC/CARC motifs for cholesterol binding. Membrane protein database analyses suggest even weaker affinity for native linear motifs, whereas live cell assays demonstrate that optimizing cholesterol binding restricts transmembrane domains to the endoplasmic reticulum post-translationally. In summary, these findings contribute to our understanding of cholesterol-protein interactions and offer insight into the mechanisms of protein-mediated cholesterol regulation within membranes.

Cholesterol serves as a major constituent of the mammalian plasma membrane. The overall fraction of cholesterol in the plasma membrane relative to total plasma membrane lipids is about 30% to 40% in leukocytes, epithelial cells, neurons, and mesenchymal cells[1]. The localization, trafficking, and functionality of membrane proteins involved in cholesterol-dependent pathways and cholesterol homeostasis may critically rely on their ability to attract and bind cholesterol molecules[2–14]. Prediction of protein-cholesterol affinity could therefore illuminate their role in diseases that are characterized by loss of cholesterol homeostasis (e.g., neurological diseases and cancer[15]), and pave the road for novel drug targets and strategies[6,11,12,16–20]. A compelling amount of data obtained by bioinformatic approaches, molecular modeling and simulations, and experiments have suggested the existence of cholesterol recognition amino acid consensus motifs

[1]Leiden Institute of Chemistry, Leiden University, Leiden, The Netherlands. [2]Technical University of Dortmund, Department of Physics, Dortmund, Germany. [3]Laboratory of Molecular Neuroscience, German Center for Neurodegenerative Diseases (DZNE), Berlin, Germany. [4]Department of Physical Chemistry, Gdańsk University of Technology, Gdańsk, Poland. [5]Department of Applied Computer Science, Gdańsk University of Technology, Gdańsk, Poland. [6]Department of NMR-based Structural Biology, Max Planck Institute for Multidisciplinary Sciences, Göttingen, Germany. [7]Cardiovascular Research Institute, University of California, San Francisco, USA. [8]Department of Pharmaceutical Chemistry, University of California, San Francisco, USA. [9]Institute of Biochemistry, Charité-Universitätsmedizin Berlin, Corporate Member of Freie Universität Berlin, Humboldt-Universität Berlin, and Berlin Institute of Health, Berlin, Germany. ✉e-mail: jelger.risselada@tu-dortmund.de

(CRAC motifs)[3,4,14,21,22], as well as its inverse (CARC motif)[23], in various membrane protein families, including, for example: viral membrane proteins (e.g., refs. 16,19), ion channels (e.g., refs. 24,25), and G protein-coupled receptors (GPCRs)—the most intensively studied drug target family (e.g., refs. 6,11,26–30).

However, the looseness of the CRAC and CARC definitions, represented via the flexible algorithmic rules: (L/V)-$X_{1-5}$-(Y)-$X_{1-5}$-(K/R) and (K/R)-$X_{1-5}$-(Y/F)-$X_{1-5}$-(L/V) respectively, is rather unexpected for a motif that mediates binding to a unique molecule, raising skepticism about its predictive value[3,10,23,31]. This flexible definition based solely on residue patterning within a single transmembrane motif neglects the overall 3-dimensional protein structure of multipass membrane proteins, including the presence of hydrophobic grooves and cavities formed between helical hairpins and additional adjacent transmembrane helices, which have been shown to actively mediate cholesterol binding[7,8,10,31,32]. In addition, the large flexibility of these motifs implies that cholesterol recognition does not depend solely on exact molecular shape compatibility, as in protein-ligand docking, but is influenced by other thermodynamic forces primarily dictated by the overall amino acid composition and structural features of transmembrane helices such as hydrophobic length and accessible surface area, similar to the structural determinants that dictate their relative preference for cholesterol-enriched membrane phases[5]. Hence, such an alternative perspective would account for the variability in the positions of these amino acids within various proposed linear motifs associated with cholesterol binding[3,14,23,33].

High-throughput screening of transmembrane sequences offers a powerful approach for investigating the existence of linear motifs while simultaneously characterizing their underlying thermodynamic driving forces. However, the accessible chemical space of transmembrane domains is astronomical (about $20^{20}$ possibilities), which warrants the use of smart search strategies.

Directed evolution is a method used in protein engineering that mimics the process of natural selection to steer proteins or nucleic acids toward a pre-specified goal[34]. Evolutionary inverse design strategies see applications in a variety of fields due to their efficient exploration of search-space[35]. These methods fall within the scope of reinforcement learning, adapting processes for optimal performance by reinforcing desired behavior[36]. Of special interest are the genetic algorithms (GA), which model the mechanisms of Darwinistic evolution in a computational algorithm, utilizing genetic elements such as recombination, cross-over, mutation, selection, and fitness[37]. Since directed evolution is both time and labor intensive, it can quickly become intractable in a laboratory setting thereby limiting its value. In such scenarios, molecular dynamics (MD) simulations may provide an alternative in silico route for the high-throughput virtual screening of chemical space.

Here, we demonstrate the ability of GAs guided by coarse-grained MD simulations—a method which we coin evolutionary molecular dynamics (Evo-MD)—to yield unique insights into the driving forces that underpin cholesterol recognition (Fig. 1). Evo-MD effectively reduces the search for optimal ligand consensus motifs to solving a variational problem in high-dimensional chemical space using stochastic operators such as genetic cross-overs and mutations. To this end, we introduce EVO-MD, a highly parallel software package for evolutionary molecular dynamics simulations that incorporates the GROMACS molecular dynamics engine into a custom, Python-based GA wrapper. EVO-MD can adapt any element of MD simulations, be it structural (e.g., atoms, molecules), topological, or simulation parameters (e.g., force field parameters), based on a reinforcement value measured during the simulation (see ref. 38 for a recent perspective on physics-based optimization).

In this work, we employ the computational method Evo-MD to explore the thermodynamic driving forces of cholesterol attraction for a fixed-length sequence of 20 amino acids within a transmembrane

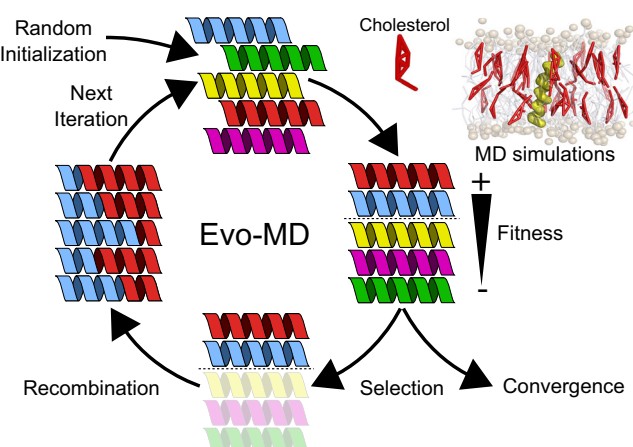

**Fig. 1 | Illustration of the basic concept of evolutionary molecular dynamics (Evo-MD).** Random peptide sequences self-evolve into optimal cholesterol attracting transmembrane domains in the course of evolution. Generated peptides are iteratively ranked upon increasing fitness, as determined via ensemble averaging within molecular dynamics simulations.

domain. Our primary objective is to investigate the factors that influence cholesterol binding affinity in transmembrane helices, guided by the hypothesis that the presence of a cholesterol-binding linear motif correlates with optimal cholesterol binding in isolated transmembrane domains. In accordance with the original linear motif concept, we exclude contributions from neighboring helices to cholesterol attraction/binding that could generate correlations extending beyond a single transmembrane domain. Our Evo-MD simulations reveal an intriguing phenomenon in this context: a strong negative hydrophobic mismatch emerges as a predominant factor in cholesterol attraction within isolated membrane helices. The resolved patterning is characterized by a short hydrophobic segment flanked by stacked charged lysine and arginine residues. This finding is further substantiated by atomistic free energy calculations, which underscore the high affinity of cholesterol for this specific hydrophobic configuration. Moreover, solid-state NMR experiments validate the interaction of cholesterol with lysine residues embedded within the hydrophobic interior of the membrane, as evidenced in synthesized transmembrane peptides. Cellular assays reveal that proteins incorporating these optimal motifs localize to the endoplasmic reticulum (ER) membrane post-translationally due to their hydrophobic mismatch.

The estimated residence time for optimal cholesterol binding is approximately hundreds of nanoseconds, which is remarkably short compared to the timescales of many biological processes. Our findings also underscore that some of the proposed essential hydrophobic aromatic residues within CARC motifs, such as phenylalanine, in fact actively and inherently repel cholesterol, refuting the prevailing assumption of their cholesterol-attracting nature. As a result, our analysis proposes that the responsiveness of specific motifs to increased cholesterol levels might be due to their use of the dual function of cholesterol as both a ligand and a solvent for membrane proteins. This responsiveness appears to rely on a fine balance between amino acids that either attract or repel cholesterol, rather than solely focusing on ligand binding.

## Results

### Cholesterol attraction features evolutionary conservation

Artificial evolution is simulated in a system consisting of a 30% cholesterol and 70% 1-palmitoyl-2-oleoyl-glycero-3-phosphocholine (POPC) membrane containing a single, 20 amino acid long peptide sequence positioned transversely through the membrane (Fig. 2A, B). The use of a model membrane composed of POPC and 30% cholesterol, though simple, effectively mimics the lipid carbon tail saturation

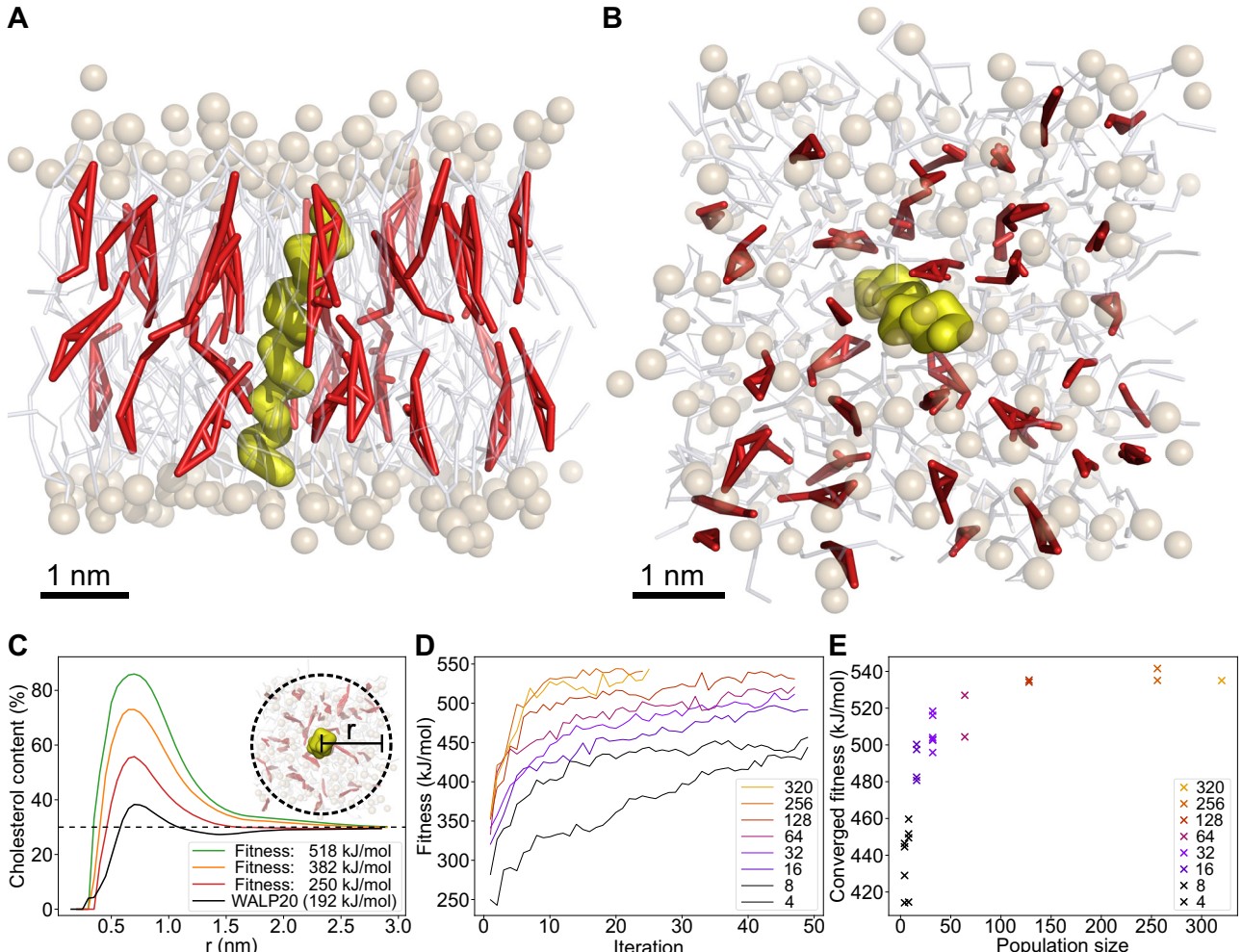

**Fig. 2 | Evolutionary molecular dynamics simulations of a cholesterol attracting transmembrane protein. A**, **B** Snapshots of a transmembrane protein (yellow) embedded within a POPC (white/brown) membrane containing 30% cholesterol (red). **C** Ratio of the cholesterol content in a local radius around the protein (see methods). An increase in fitness correlates to an increase in local cholesterol. The baseline cholesterol concentration (30%) is indicated by the dashed line. **D** Fitness development during protein evolution, shown for various population sizes. The fitness is expressed in terms of the total peptide-cholesterol non-bonded interaction energy. Fitness increases with GA iterations. Size of the population affects the height of the fitness plateau. **E** The GA converges to different fitness values, depending on the size of the populations. Eventually, evolution converges to an optimal solution for population sizes greater than 128 individuals.

and cholesterol concentration found in many cellular membranes[39]. We conducted simulations using both the Martini 2[40–42] and the newer Martini 3[43–45] coarse-grained force fields to validate for potential inconsistencies between the force fields. Owing to the symmetry of the here studied bilayer, generated sequences are mirror symmetric, i.e., only the first ten amino acids are independently chosen. Evolution is directed towards peptide sequences that increase the local density of cholesterol, visualized by the percentage cholesterol content of the membrane within a certain range from the peptide (Fig. 2C). In practice, this is obtained by maximizing the ensemble-averaged non-bonded interaction energy between the peptide and cholesterol, i.e., this defines the fitness, in the course of sequence evolution.

Starting from random peptide sequences, the observed evolution eventually converges to an optimum, as is evident by a plateau in the fitness values (Fig. 2D). Convergence of genetic algorithms depends on a variety of factors, most notably the size of the population—which directly relates to the area of the search space that is sampled each iteration—and the number of iterations that are performed. Either parameter requires some minimum value for convergence to occur. The population size should be large enough (in combination with mutation rate and other diversifying factors) to prevent premature convergence to suboptimal solutions, and, with evolution proceeding

between iterations, a certain minimum number of iterations is necessary. Ideally, both parameters are chosen as large as possible.

To assess whether the convergence of evolution is either suboptimal (i.e., a local solution) or optimal (i.e., a global solution), we conducted a set of evolutionary runs with population sizes ranging from 4 to 320 individuals until no further convergence of fitness was observed. Figure 2D shows how the fitness of the best-performing sequences changes with each generation. As expected, increasing population size increases the optimum fitness, as is evident from a higher plateau value reached after convergence of fitness (Fig. 2E). This increase in optimal fitness leveled off once the population size began exceeding 128 individuals, which we took as the baseline population size for GA convergence. Data from GA runs containing 128+ individuals and at least 40 generations was used for sequence analysis.

Associated with the convergence in fitness with respect to population size, we observed a similarity in the sequences produced by distinct GA runs. Although GA runs with smaller population sizes (<64) eventually converged to some fitness value, a comparison between these distinct GA runs revealed a large diversity in the respective sequences, indicating that the algorithms converged to local optima in the solution space. This diversity in sequence decreases as population size increases, with very similar sequences being obtained as

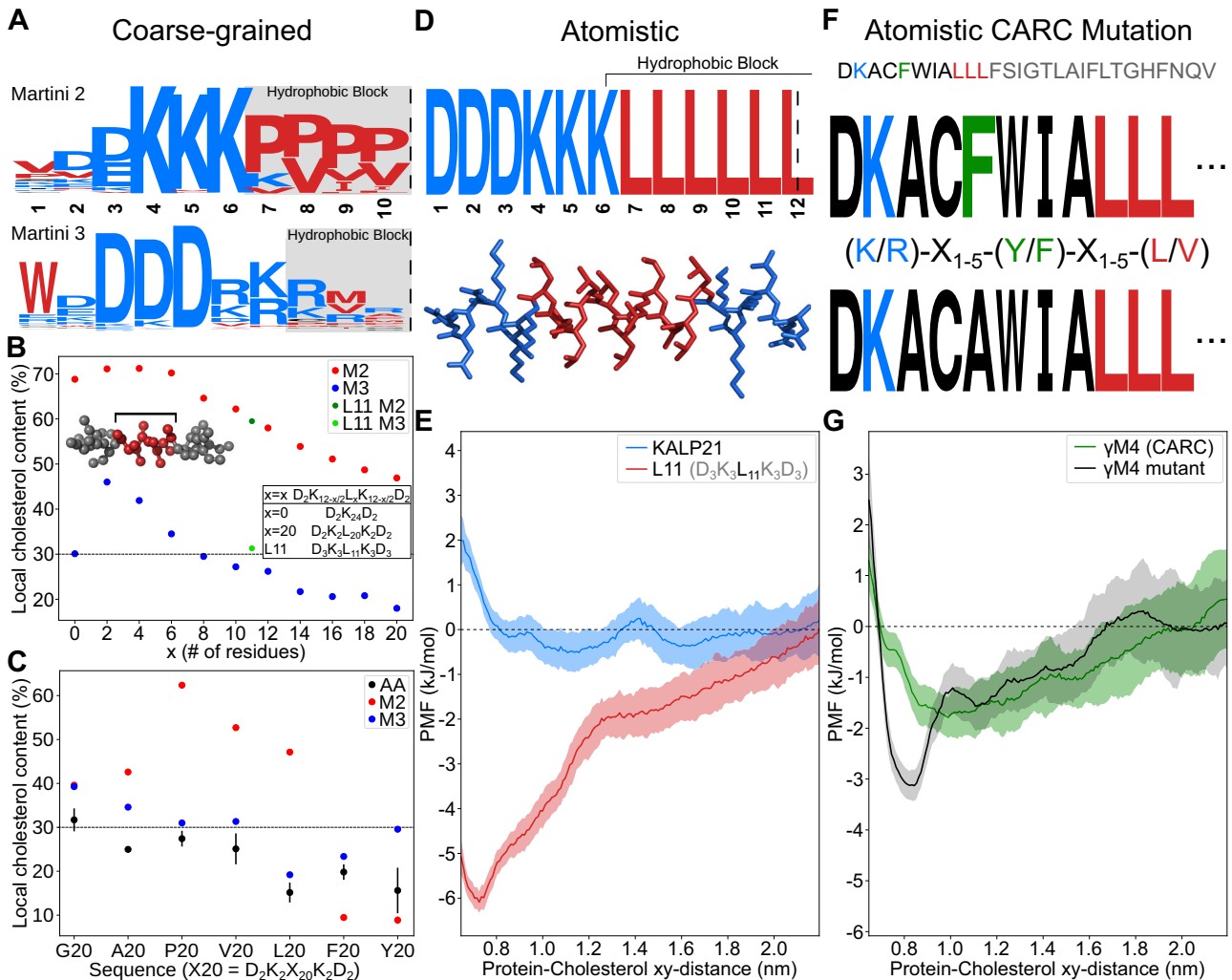

**Fig. 3 | Sequence and chemical features of the optimal cholesterol attractor.**
**A** Sequence logos computed from high-fitness peptide sequences reveal a highly conserved hydrophobic mismatch pattern (red = hydrophobic; blue = hydrophilic). Owing to the symmetry of the here-used bilayer, sequences are mirrored around the center as indicated by the dashed line. **B** Peptide-induced hydrophobic mismatch leads to a high local (1.0 nm radius) cholesterol composition of the membrane. This mismatch mechanism is present in both Martini 2 and Martini 3. Sequences adhere to the following motif: $D_2K_{(12-x/2)}$--$L_x$--$K_{(12-x/2)}D_2$ ($x = 0, 2, 4$ etc.). **C** Analysis of side-chain cholesterol affinity across force fields. Martini 2 shows a preference for small hydrophobic side-chains (P, V, L, A). This mechanism is absent in Martini 3 and all-atom, where the emphasis seems to lie on the size of the sidechain. Error bars represent the standard error of the mean. Statistics were obtained from 3 independent replicates. **D** A rationally designed motif (L11) based on the CG optimal cholesterol attractor. The sequence retains both the conserved poly-lysine patches, and the short hydrophobic section. The corresponding

atomistic structure is shown below. **E** Free energy profiles over the peptide-cholesterol distance are computed in all-atom simulations for the rationally designed motif $D_3K_3L_{11}K_3D_3$ (L11), and the stereotypical transmembrane peptide $GK_2[LA]_7LK_2A$ (KALP21). KALP21 is characterized by a slender hydrophobic motif rich in leucines $[LA]_7L$. Nevertheless, a pronounced cholesterol affinity is only observed for the designed motif L11. Shaded areas represent the standard error of the mean. 3 independent replicates were simulated for each peptide. **F** Peptide covering the known cholesterol binding $\gamma$ M4 transmembrane region[4,56]. The CARC motif present in this sequence is indicated by the colors[4,56]. Mutation of the aromatic residue phenylalanine into an alanine is known to impair its cholesterol-dependence[4]. **G** Free energy profiles over the peptide-cholesterol distance are computed in all-atom simulations for the $\gamma$ M4 peptide, and the non-CARC (F → A) mutant of the $\gamma$ M4 peptide. Mutation of phenylalanine in fact produces a strong increase in cholesterol affinity. Shaded areas represent the standard error of the mean. 3 independent replicates were simulated for each peptide.

population sizes increase to 128 individuals and above. Furthermore, at such population sizes, starting the evolution from different initial populations consisting of randomly generated sequences yields a consistent result. On these grounds, we can conclude that the GA successfully converges to a global optimum.

To gain detailed insights into the resolved evolutionary landscape, high-fitness sequences from all GA runs with populations of 128+ individuals were combined to generate a sequence logo of the sampled sequence space (Fig. 3A). Sequence logos express the degree of amino acid conservation at each position within the sequence in terms of the concomitant Shannon entropy (bits) by scaling the character height of the corresponding amino acid. Randomly

occurring amino acids at a certain position contain no information, corresponding to a small letter, whereas a more frequently occurring amino acid encodes information, corresponding to a larger letter.

In both the Martini 2 and Martini 3 coarse-grained force fields, the global solution converges to a distinctive pattern featuring a short conserved hydrophobic core centered within the peptide. This core is flanked by two hydrophilic blocks composed of conserved positively charged lysines (K) and arginines (R). Notable differences exist between force fields. Martini 2 exhibits a strong preference for three consecutive lysines, which are the most evolutionarily conserved residues. In contrast, Martini 3 features equal competition between lysines and arginines. Both versions primarily feature negatively

charged aspartic acids (D) at terminal positions, which are more highly conserved in the Martini 3 force field. High-fitness sequences resulting from directed evolution in both Martini 2 and Martini 3 force field versions exhibit a consistent hydrophobic pattern. This pattern features positively charged lysines and arginines at positions directly facing a central short hydrophobic block.

It is important to emphasize that the solution space resolved here is subject to a constraint in secondary structure, i.e., all sequences are assumed to be alpha-helical[5]. We will address the transferability of solution space in more detail in a later section of this work. Furthermore, while our study primarily investigates cholesterol attraction in simplified POPC model membranes, it is important to note that verification using a coarse-grained model of native epithelial membrane[46] demonstrates the universality and persistence of the resolved attraction features in more realistic membrane environments (Supplementary Fig. 2).

### Short hydrophobic blocks maximize cholesterol attraction

The sharp positional convergence of hydrophilic charged residues deeply located in the hydrophobic core of the membrane prompted us to investigate what role the length of the hydrophobic block plays in the cholesterol-sensing ability of the sequence. To this end, we created dummy peptides according to the $D_2K_{(12-x/2)}-L_x-K_{(12-x/2)}D_2$ motif with each peptide consisting of 20 amino acids in total. Here, leucines form the hydrophobic block of the peptides, with lysines functioning as the hydrophilic edges. By varying the number of leucines and lysines, we effectively vary the length of the hydrophobic block. Interestingly, cholesterol affinity increases with decreasing hydrophobic block length, with an optimal effect at 2–4 leucines (Fig. 3B). This pattern seems to arise from a trade-off between short block length and transmembrane (meta)stability, with a further decrease in block length resulting in a decline in functionality. Artificially restraining a transmembrane orientation/topology for such motifs (e.g., $K_9V_2K_9$, and even $K_{20}$) eliminates the stability factor, thereby restoring the functionality (Supplementary Fig. 3). The cholesterol attraction thus appears to be mediated by positively charged lysine residues deeply embedded in the membrane, as is consistent with their evolutionary conservation. The positioning of these residues, specifically the length of the conserved hydrophobic block, must ensure a transmembrane topology during evolutionary development. Interestingly, despite the Martini 2 force field showing a stronger net attraction than the Martini 3 force field, both exhibit a similar overall gradual decline in relative cholesterol affinity as block size increases toward a hydrophobic length of 20 amino acids.

Finally, we emphasize that our study specifically focuses on maximizing the attraction of free membrane cholesterol. Owing to the membrane thickening effect of cholesterol[42], cholesterol-enriched phases such as the liquid ordered (Lo) phase generally favor TMDs characterized by a long rather than short hydrophobic length[5,47–50]. The here-resolved motif is therefore not expected to optimally bind toward the interface of cholesterol-enriched liquid ordered domains[5,42] (Supplementary Fig. 9). Nevertheless, the clustering of cholesterol is itself membrane phase independent and equally occurs when the resolved TMD is embedded within a liquid-ordered DPPC:cholesterol mixture (Supplementary Fig. 3).

### Cholesterol affinity favors small hydrophobic amino acids

Next, we examined whether the composition of the hydrophobic fraction influences cholesterol attraction. To investigate this, we constructed $D_2K_2X_{20}K_2D_2$ sequences to systematically analyze the native cholesterol affinity of hydrophobic residues in the absence of hydrophobic mismatch for the Martini 2, Martini 3, and the atomistic (AMBER99SB-ILDN with Slipids) force field (Fig. 3C). We measured the local cholesterol composition within a 1.0 nm radius of the transmembrane domain to assess cholesterol attraction.

In the Martini 2 force field, we observed an unexpectedly strong attraction between cholesterol and certain amino acid residues, particularly proline, valine, and leucine. These residues are modeled using a simplified representation consisting of a small single-bead side chain with variable bond lengths. Our investigation revealed that artificially altering the side chain bond distances significantly impacted cholesterol attraction. Specifically, decreasing the bond length enhanced cholesterol attraction, while increasing it diminished attraction (Supplementary Fig. 20). We attribute this pronounced cholesterol attraction primarily to artifacts arising from the exaggerated interactions between small bead types used to represent both cholesterol and amino acids within the Martini 2 force field[43,44].

In contrast, the Martini 3 force field showed a different pattern. Only alanine and glycine displayed significant net attraction toward cholesterol. However, the atomistic simulations revealed that only glycine may exhibit a weak but significant cholesterol attraction. This finding aligns with the cholesterol binding to glycine zipper motifs observed in atomistic simulations[17,51].

Surprisingly, larger hydrophobic aromatic residues such as tyrosine (Y) and phenylalanine (F)—key components of CRAC/CARC motifs—were found to be weakly or strongly cholesterol repulsive across all simulation models, including atomistic simulations. Furthermore, other hydrophobic CRAC/CARC residues like leucine and valine showed either inert or repulsive behavior toward cholesterol, with particularly strong repulsion observed in the atomistic simulations.

Our research across three distinct force fields reveals that the composition of hydrophobic residues may prioritize minimizing cholesterol repulsion over maximizing attraction. Notably, the atomistic and Martini 3 force fields demonstrated greater behavioral similarity compared to the Martini 2 force field. To minimize repulsion, simulations consistently favored small hydrophobic amino acids, such as alanine, and residues with weaker helical propensity, including valine, proline, and glycine. Conversely, larger hydrophobic amino acids like leucine and aromatic amino acids (phenylalanine and tyrosine) enhance repulsion. This pattern suggests that cholesterol affinity appears more dependent on the size rather than the hydrophobicity of the hydrophobic amino acids constituting transmembrane helices. We propose that bulky, highly corrugated proteins disrupt the order within the surrounding cholesterol matrix[5,52], resulting in a local depletion of cholesterol. The surprising absence of correlation between amino acid hydrophobicity and (relative) cholesterol affinity suggests that depletion is likely driven by optimizing cholesterol-cholesterol interactions rather than protein-cholesterol interactions. Hydrophobic transmembrane domains therefore tend to show a net repulsion rather than a net attraction toward cholesterol. This repulsion appears to be compensated by negative hydrophobic mismatch via lysines and arginines exposed to the hydrophobic membrane core.

### NMR experiments and Atomistic MD support the resolved motif

In this work, we resolved the essential physicochemical driving forces that underpin cholesterol attraction in transmembrane domains within homogeneous model membranes. The here-resolved motif features of the optimal cholesterol attractor are subsequently translated into more realistic peptide sequences by accounting for the following three model approximations:

(I) Given that transmembrane domains are primarily composed of alpha-helices, we imposed an alpha-helical secondary structure constraint on the generated sequences. Although this assumption simplifies the search space by bypassing the challenge of secondary structure prediction, it introduces the potential for amino acids with low alpha-helix propensities (such as proline and valine)[53] to appear in the generated sequences, potentially leading to non-helical peptides in unconstrained simulations. Maintaining stable helicity is crucial for preserving membrane stability. Short hydrophobic helical segments

flanked by deeply embedded charged amino acids create negative hydrophobic mismatch, which maximizes cholesterol attraction. However, strong membrane elastic forces constantly counteract this stability. To address this, we designed a polyleucine sequence due to its high helical propensity. Note that leucine residues exhibit inherent cholesterol repulsion in our atomic-scale simulations (Fig. 3C).

(II) Electrostatic interactions are underestimated in the coarse-grained simulations, enabling the formation of sequences with a high net charge. To obtain a sequence with net zero charge, we balance the conserved lysines patches by adding three aspartic acids (D) to both terminal ends. This essentially entails a superposition of the conserved features observed in the Martini 2 and 3 force fields.

(III) We anticipate on the notion that the coarse-grained model—and MD simulations in general—underestimate the hydrophobic length where transmembrane domains become thermodynamically stable with respect to experimental conditions. Transmembrane partitioning of polyleucine helices in experiments only becomes favorable over surface partitioning at a length of 10 leucines, in contrast to their atomistic estimation of 7–8 leucines[54] and our course-grained estimation of 6 leucines (Supplementary Fig. 4).

Altogether, this leads to the more realistic sequences $D_3K_3L_{11}K_3D_3$ (L11) and potentially $D_3K_3L_{10}K_3D_3$ (L10), both of which retain all the design features proposed by the GA. Biophysical characterization in model membranes (POPC and DLPC with 30% cholesterol) using Circular Dichroism (CD) spectra confirms that even the shorter of these two sequences (L10) adopts a helical structure in lipid membranes (Supplementary Fig. 15).

The L10 peptide was confirmed to associate with cholesterol through its highly conserved lysine patch in NMR experiments that correlate peptide and cholesterol signals when the two are in close contact. The peptide was labeled with $^{13}C$ at the carbonyl group of the two lysine residues that are directly adjacent to the Leucine motif (position 6 and 17 in the sequence) and $^{13}C4$ labeled cholesterol. The use of ether-linked lipids avoids any signal in the carbonyl region coming from the lipids. A cross peak in the PDSD spectrum (Fig. 4A, Supplementary Fig. 5) between the carbonyl group and cholesterol C4 confirms the interaction. A comparable interaction is seen for KALP-21 (Supplementary Fig. 6), which was labeled at the analogous lysine residues. For these measurements, the sample temperature was 100 K to prevent diffusion, allowing a long mixing time of 30 s, which is needed to efficiently observe transfer over the expected distance range of about 6 to 9 Å[55]. Note that the transfer rate in PDSD is expected to scale down with the sixth power of distance, such that the measurement is strongly influenced by any small changes in the pose of the cholesterol molecule relative to the peptide (See Fig. 4 for a depiction of two such poses of close contact between peptide and cholesterol, in which the distance changes substantially).

Furthermore, free energy calculations in atomistic MD simulations (see *Methods*) confirm that this design pattern exhibits a pronounced functionality in cholesterol affinity, as shown in Fig. 3E for the sequence L11. This functionality is particularly evident when compared to (i) the prototypical and somewhat similar model peptide KALP21 (sequence: GKK(LA)$_7$LKKA), (ii) the $\gamma$M4 transmembrane domain of the muscle nicotinic acetylcholine receptor—a known strong cholesterol binding sequence with a CARC motif[4,56], and (iii) its F-452/A mutant[4] (see Fig. 3G and Supplementary Fig. 21). We thus observe that the encoded functionality persists between the different model resolutions. Moreover, the obtained free energy profile illustrates that cholesterol attraction occurs over rather large distances—up to 1.8 nm—suggesting that the attraction is membrane mediated, and thus resulting from an interplay between peptide and membrane.

Optimization of cholesterol binding resulted in a thermodynamic optimum characterized by a small free energy minimum of up to 5 kJ/mol or 2 $k_BT$. Notably, this optimum represents the upper limit of achievable residence time for optimal cholesterol binding. To put such

a value into perspective: The binding free energy of typical ligands modifying GPCR function exceeds values of 40 kJ/mol or 16 $k_BT$[57] and is thus substantially larger than that of cholesterol acting as a ligand via binding of linear motifs.

Notably, our fitness function effectively maximizes the integral of the free energy profiles shown in Fig. 3 within the cutoff radius of the simulation (1.2 nm). To elucidate its association with the maximum binding affinity of a single cholesterol molecule, we analyzed multiple sequences, including the well-established CRAC and CARC motifs. An overview of the measured fitness and the associated (maximum) binding affinity of a single cholesterol molecule is listed in Supplementary Fig. 17. The linear correlation we observed provides evidence for the correlation between the maximum binding affinity of a single cholesterol molecule and the overall enthalpic interaction. Therefore, optimizing the attraction between cholesterol and the membrane environment simultaneously optimizes the binding affinity for individual cholesterol molecules, and thus we observed the upper thermodynamic limit of cholesterol binding to linear motifs.

Our analysis of the concomitant average first passage times (see Methods), derived from atomistic simulations, reveals that the upper bound for cholesterol-binding residence time falls below 400 ns for the L11 sequence. Although linear motifs within transmembrane domains can facilitate cholesterol binding, the low binding affinity and concomitant short residence time—even when close to the thermodynamic optimum—may significantly limit the ability of such a ligand binding based mechanism to alter protein functionality within GPCRs, given that concomitant changes within the conformational ensemble due to ligand binding occur on microseconds to milliseconds time scales[58,59].

## The mechanism behind optimal cholesterol attraction

The main question to address remains why the thermodynamically optimal mechanism of cholesterol attraction favors hydrophobic mismatch. Notably, the observed effect is consistent across different force fields, demonstrating robustness and reliability. Specifically, the phenomenon occurs in all three force fields tested, suggesting that the underlying physical principles driving this behavior are not dependent on the particular set of parameters used in molecular simulations. In contrast to POPC lipids, cholesterol exhibits a low free energy barrier when undergoing flip-flopping between the two leaflets of the membrane. As a result, the head group of cholesterol is particularly adept at interacting with the lysines deeply located within the hydrophobic region of the membrane. Such binding mode is confirmed both by our molecular dynamics simulation as well as solid state NMR experiments (Fig. 4A). We hypothesize that by moving toward this hydrophobic region, cholesterol molecules effectively shield the lysine patch from unfavorable interactions with the hydrophobic lipid tails (Fig. 4B). To this end, we conducted simulations within the Martini 2 force field that artificially restricted bilayer flip-flopping of cholesterol in the simulations via the application of an external field (flat-bottom potentials). High-fitness sequences containing a short hydrophobic block, which would rely on the vertical mobility of cholesterol for their functionality, experienced a significant decrease in cholesterol attraction. However, longer attractors with less optimal characteristics, where the attraction of cholesterol primarily depends on the nature of the hydrophobic section, remained relatively unaffected (Fig. 4C). Therefore, we attribute the enhanced attraction of cholesterol to the difference in vertical mobility of lipid head groups in the immediate vicinity of the transmembrane domain (TMD). It is worth noting that the thermodynamically optimal POPE attractor (Martini 2 force field) can also be attributed to a differential vertical mobility effect between POPE and POPC lipids due to the effectively smaller phosphatidylethanolamine (PE) head group. However, in this case the attractors exploit a favorable enthalpic interaction between POPE head groups and the centrally located tryptophan region (Supplementary Fig. 14).

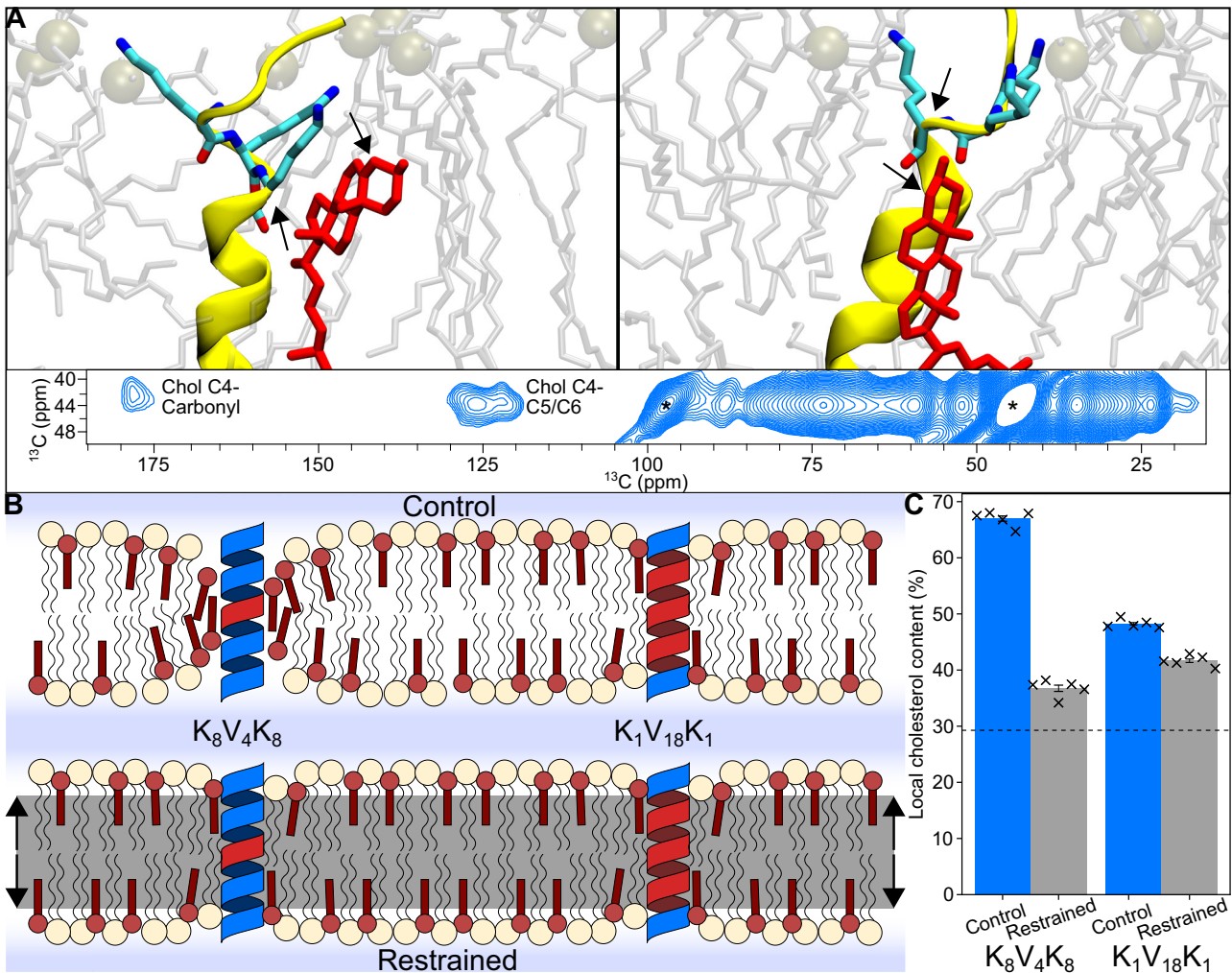

**Fig. 4 | Mechanistic model for optimal cholesterol attraction. A** All-atom MD snapshots display peptide-cholesterol interaction deep within the membrane. Interaction between cholesterol (C-4) and the deeply located lysine residues (carbonyl of position 6 and its mirror in the sequence logo) is also observed in DNP-enhanced ssNMR (inset). Labeled lysines and the cholesterol C-4's are indicated with black arrows. **B** (*Control*) Cholesterol (red) exhibits a lower energy penalty for movement along the membrane normal compared to POPC (beige), allowing for more favorable shielding of deeply located lysine residues. High-fitness cholesterol attractors utilize this effect by increasing deep lysine interactions, leading to local accumulation of cholesterol molecules. **B** (*Restrained*) Application of a force to lipid headgroups within a specific distance from the membrane cter prevents cholesterol flip-flopping and movement toward the bilayer center. **C** Removal of cholesterol vertical mobility (Restrained) leads to a large drop in functionality for cholesterol attractors with short hydrophobic blocks ($K_8V_4K_8$), while attractors with longer hydrophobic blocks ($K_1V_{18}K_1$) are less affected. The dashed line indicates the average cholesterol content of the system (30%). Bars represent the standard error of the mean. Statistics were obtained from 5 independent replicates.

## Exploitation of hydrophobic mismatches is limited by nature

An interesting question is to what extent a hydrophobic mismatch mediated attraction of cholesterol can be expressed within isolated transmembrane domains in nature. Noting that hydrophobic mismatch is also a known determinant in protein trafficking and sorting[49,60], one would therefore intuitively expect a stronger limitation on the evolutionary expression of such a mechanism. To investigate the possible nature of these evolutionary constraints, we performed experiments in live cells (HEK cells) expressing the short hydrophobic sequences $D_3K_3L_{10}K_3D_3$ (L10) and $D_3K_3L_{11}K_3D_3$ (L11), each with a fluorescent tag, as well as KALP21 (GK$_2$[LA]$_7$LK$_2$A). KALP21 is a typical model peptide in membrane biophysical studies and has a (relatively short) hydrophobic length of 15 amino acids. Our experiments revealed that L10 (Fig. 5E), L11 (Fig. 5F), and KALP21 (Fig. 5G) can be effectively expressed in live cells. These transmembrane proteins were found to localize exclusively to the endoplasmic reticulum (ER) and not to other intracellular organelles such as lysosomes or mitochondria (Supplementary Fig. 10). In addition,

they did not localize to the plasma membrane, but notably decreased the trafficking of fat transporter and scavenger receptor CD36 to the plasma membrane (Supplementary Figs. 11 and 13). The unique characteristics of the ER membrane make it particularly favorable for the insertion of transmembrane domains (TMDs) with negative hydrophobic mismatch, as it is the thinnest membrane in live cells and incurs the lowest energetic penalty for such insertions[60,61]. In contrast, the TMDs of SNARE proteins (such as Syntaxin-1), which have longer hydrophobic lengths ranging from 23 to 25 amino acids, can still be successfully expressed throughout the cell using the assay employed in this study[49]. However, the fact that a prototypical model peptide like KALP21 (with a hydrophobic length of 15 amino acids), which differs by only one amino acid from the shortest native TMD within the TmAlphaFold database (with a hydrophobic length of 16 amino acids), is confined to the ER membrane highlights the existence of an evolutionary barrier related to protein trafficking. This barrier prevents optimal exploitation of the hydrophobic mismatch mechanism, which favors a hydrophobic

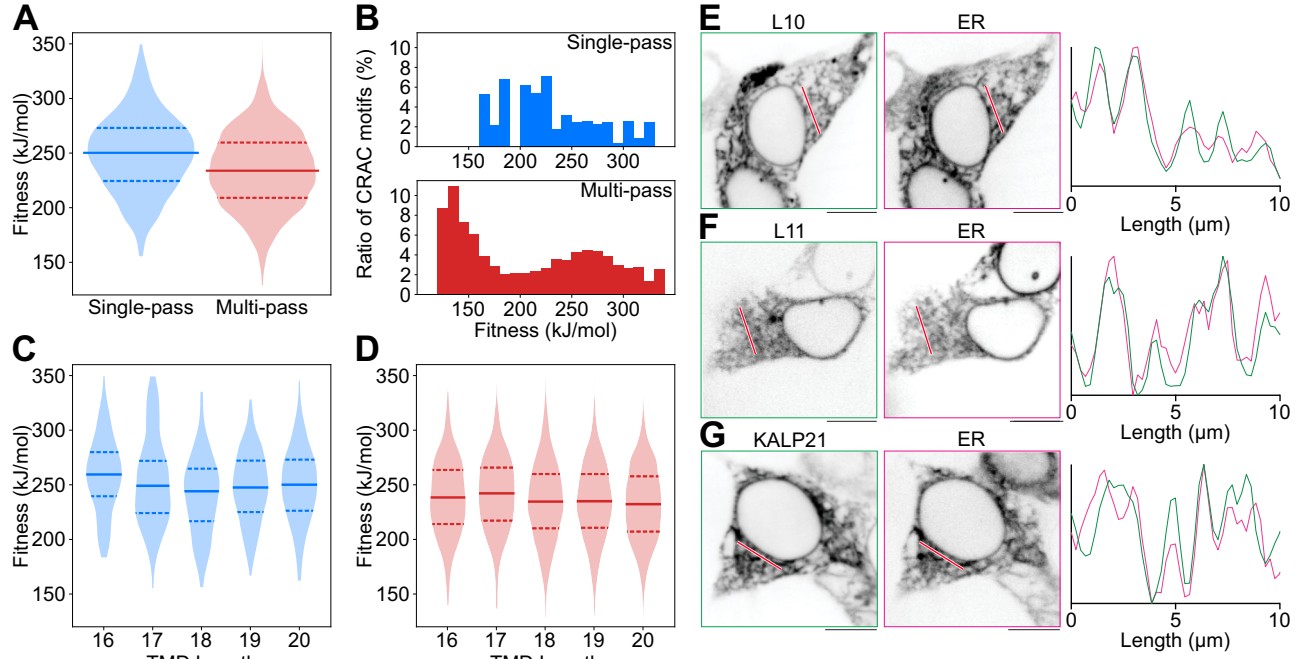

**Fig. 5 | Existence and viability of mismatch-based attraction in nature: an analysis of the TmAlphaFold Transmembrane Protein Structure Database[78].** **A** Comparison of CNN-predicted fitness distributions between single-pass and multi-pass database TMDs. Markers indicate the interquartile range and the median of the data. **B** Relative presence of CRAC motifs ((L/V)-X1-5-(Y)-X1-5-(K/R), and its inverse) with respect to the CNN-predicted fitness. **C** Single-pass CNN-predicted fitness distributions with respect to TMD length. Markers indicate the interquartile range and the median of the data. **D** Multi-pass CNN-predicted fitness distributions with respect to TMD length. Markers indicate the interquartile range and the median of the data. Fluorescence microscopy of transfected HEK cells, expressing L10 (**E**), L11 (**F**), and KALP21 (**G**); as well as fluorophore-tagged Sec61 to mark the ER. For each panel, a line profile was drawn (red), and the normalized fluorescence intensity profiles of peptide and ER are compared in the respective graphs. Scale bars and line profiles in all panels correspond to 10 μm. All microscopy experiments were performed in three independent replicates.

length toward the limit of transmembrane topology stability (10 to 11 amino acids).

Finally, to explore the potential exploitation of hydrophobic mismatch-mediated attraction in nature, we systematically analyzed isolated transmembrane domains extracted from 8370 native membrane proteins in the TmAlphaFold database using a CNN trained on EVO-MD fitness-labeled data within the Martini 2 model (see Methods and Supplementary Information). We discovered a weak but significant correlation between predicted fitness and TMD length in single-pass proteins, specifically at the shortest hydrophobic length of 16 amino acids (Fig. 5C). This correlation was absent in multi-pass proteins, likely due to differential TMD lengths diminishing weak evolutionary pressures for the expression of negative hydrophobic mismatch. As a result, cholesterol attraction via linear motifs in nature will be limited toward less efficient mechanisms yielding residence times that likely fall below the timescale of several 100 ns estimated for optimal cholesterol attraction/binding. This raises the question whether these thermodynamically suboptimal mechanisms could remain effective in achieving their biological purpose, specifically the regulation of GPCRs, given that the timescales of conformational responses (relaxation times) within GPCRs upon binding of ligands, being high microsecond to milliseconds[58,59], lie far beyond the here estimated range of maximal attainable residence times for cholesterol binding to linear motifs.

## CRAC/CARC seems not predictive for cholesterol attraction

The CRAC/CARC motif has traditionally served as the primary criterion for predicting cholesterol attraction/binding within transmembrane domains (TMDs). However, our study aimed to reassess this motif's predictive capacity for accurately determining cholesterol attraction, its proposed functional role. Interestingly, our Evo-

MD simulations revealed that aromatic residues crucial for the CRAC/CARC motif were not conserved during the evolutionary process aimed at optimizing cholesterol attraction. In addition, systematic atomistic simulations demonstrated that hydrophobic motifs consisting of the aromatic CRAC/CARC residues F and Y strongly repel cholesterol. The most potent cholesterol binding motif described in the scientific literature, as revealed through in silico molecular docking, is a CARC motif found within the γ M4 transmembrane domain of the muscle nicotinic acetylcholine receptor[4,56]. Although our atomistic simulations confirmed a modest initial affinity for cholesterol, as indicated by a shallow free energy minimum of approximately 2.3 kJ/mol, the introduction of a putative mutation (F-452/A) in the crucial aromatic residue within the CARC motif, replacing phenylalanine with alanine[4], actually enhanced the motif's ability to attract cholesterol rather than impairing it (Fig. 3G). This finding is consistent with the detrimental effect of phenylalanine on cholesterol attraction when it forms the hydrophobic motif, as observed in our coarse-grained and atomistic simulations. Notably, the characteristic free energy well depth for cholesterol attraction in linear motifs is small (on the order of $k_BT$), leading to considerable variations between replicas in individual umbrella sampling attempts (Supplementary Fig. 21). However, the free energy differences between the different peptide sequences, particularly between L11, KALP21, and γ M4, are pronounced and substantially larger than the sampling noise.

Our results challenge the current assumption of CRAC/CARC motif functionality in transmembrane domains (TMDs), as the presence of hydrophobic CRAC/CARC residues V, L, F, and Y within hydrophobic motifs−being larger amino acids−intrinsically decreases rather than increases cholesterol attraction (also see Supplementary Fig. 12). The discrepancy between observed behavior and proposed

roles in optimizing cholesterol binding affinity raises questions about the true biological functions of these motifs. Key observations include:

- Short residence times: The cholesterol binding free energy to CRAC/CARC motifs is $2\,k_BT$ or less (<5 kJ/mol), indicating low affinity when compared to the energy of thermal fluctuations. Consequently, the residence time is only several hundred nanoseconds, suggesting rapid dissociation compared to other ligands known to regulate GPCRs[57].
- Cholesterol repulsion: Key residues in CRAC/CARC motifs repel cholesterol, contradicting expectations based on their proposed function.
- Absence of co-crystal structures: No crystal or Cryo-Electron Microscopy (Cryo-EM) structures feature cholesterol bound to CRAC/CARC motifs, suggesting generally weak binding affinities[10,31]. Despite molecular docking studies in vacuum showing good fit for cholesterol binding to CRAC/CARC motifs[3,23], actual experimental support for this interaction remains scarce.

Finally, we conducted a comprehensive analysis using the TmAlphaFold database for membrane proteins, employing a convolutional neural network (CNN) trained on fitness-labeled data generated by EVO-MD using the Martini 2 model (see Methods and Supplementary Information). The Martini 2 coarse-grained force field has been successfully applied in modeling cholesterol binding to CRAC motifs present in serotonin1A receptors and ErbB2 growth factor receptors 2[27,62]. Although this coarse-grained model systematically overestimates cholesterol attraction in transmembrane proteins[45], its behavior aligns with the atomistic simulations regarding cholesterol repulsion by aromatic residues. The performed analysis examined the frequency of CRAC/CARC motifs and their correlation with cholesterol attraction (Fig. 5B). Our findings reveal a negative correlation between cholesterol attraction and the occurrence of CRAC motifs in both single-pass and multi-pass transmembrane domains (TMDs). Notably, systematic mutation of these residues to alanine significantly increases cholesterol attraction (Supplementary Fig. 12). This phenomenon can be attributed to the cholesterol-repulsive nature of hydrophobic aromatic residues required to classify a motif as CRAC/CARC.

These observations collectively indicate that mechanisms governing CRAC/CARC motif function in TMDs may differ significantly from their proposed role in optimizing cholesterol binding affinity. This conclusion highlights the need for further research to elucidate the potential recognition mechanisms of linear motifs. Specifically, it emphasizes the need for further investigation of examples where point mutations in identified CRAC/CARC motifs have impaired cholesterol responsiveness, such as the motifs present in the Programmed death-ligand 1 (PD-L1) and serotonin 1A receptor, or the mitochondrial translocator protein TSPO[11–13].

## Discussion
Our study applied Evo-MD simulations to investigate the mechanisms and design features responsible for driving optimal cholesterol attraction within transmembrane domains (TMDs). We found that hydrophobic mismatch and the presence of small hydrophobic amino acids play significant roles in facilitating the ideal interaction between cholesterol and TMDs. These mechanisms demonstrated robustness across multiple simulation models, diverse simulated membrane compositions (including a coarse-grained model of the native epithelial membrane[46]), as depicted in Supplementary Fig. 2, and various membrane environments such as the liquid-disordered and liquid-ordered phases, as shown in Supplementary Fig. 9. These findings emphasize the fundamental importance of these mechanisms in governing cholesterol-membrane interactions within native membrane proteins.

In the field of cholesterol-binding domains, the CRAC (cholesterol recognition/interaction amino acid consensus) and its inverse motif

CARC have gained significant attention and are widely studied in scientific literature. These motifs have been identified in various proteins known to interact with cholesterol, particularly GPCRs (G-protein coupled receptors)[3]. However, there is an ongoing debate regarding the applicability of CRAC/CARC motifs in GPCRs. It has been observed that cholesterol can crystallize bound to GPCRs that lack a CRAC, CARC, or the equivalent cholesterol consensus motif (CCM) that switches the position of Y/F residue and L/V within the CRAC algorithm[30,33,63], and even when these motifs are present, cholesterol often does not occupy them[10,14,31,64]. This highlights the complexity of cholesterol-protein binding and suggests that additional mechanisms beyond CRAC/CARC motifs may contribute to cholesterol binding in GPCRs. Our study adds to this understanding by exploring the broader mechanisms and design features that govern cholesterol attraction in linear motifs. We demonstrated that isolated transmembrane domains can facilitate cholesterol binding akin to the concept of linear motifs, albeit with very low affinity (up to $2\,k_BT$) and short residence time (up to 400 ns) even in the thermodynamic optimum.

Previous atomistic simulations have explored how cholesterol modulates the human $\beta$2-adrenergic receptor ($\beta$2AR), a prototype G protein-coupled receptor, in an allosteric manner[63]. The proposed mechanism involves cholesterol binding to specific high-affinity sites near transmembrane helices 5-7 of the receptor. Notably, the lifetime of cholesterol in these high-affinity sites was found to be (at least) microsecond-scale, thus significantly longer than the nanosecond lifetimes observed for linear motifs.

The binding of typical regulatory ligands targeting GPCRs is 42 kJ/mol (10 kcal/mol) or about $16\,k_BT$[57] and exceeds the here measured binding free energy of cholesterol to optimal linear motifs (about $2\,k_BT$) by about $14\,k_BT$[57]. This would therefore result in a concomitant residence time that is, assuming a similar kinetic prefactor, $1.2 \times 10^6$ times longer−thus approaching second time scales. It can be argued that, due to the high abundance of cholesterol within the plasma membrane, the binding occupancy will be high despite weak binding interactions. Nevertheless, it remains questionable whether the rapid ligand binding and unbinding kinetics associated with linear motifs can sufficiently influence the slower relaxation modes within membrane proteins, which are relevant for functionality and occur on and above microsecond timescales[65].

Aromatic residues are considered the key components in the CRAC, CARC, and CCM motifs[3,23,33]. Notably, the contribution of aromatic residues to the binding affinity within CARC/CARC motifs has been primarily inferred from the enthalpic interactions observed in docking experiments with a single cholesterol molecule in a vacuum[3]. Our simulations sought to replicate and extend these findings by maximizing enthalpic interactions within a more realistic membrane environment. In such an environment, the interactions with phospholipids become competitive since the attraction of cholesterol is mediated by relative differences in binding affinity with other lipids, rather than relying solely on absolute cholesterol binding affinity as measured within in vacuo docking experiments.

Having shown that hydrophobic aromatic residues tend to be detrimental to cholesterol attraction in isolated linear motifs within a lipid environment, the following question emerges: is their presence coincidental, arising from other evolutionary pressures unrelated to cholesterol-mediated regulation of transmembrane proteins (such as structural stability or the decreased packing of lipids in membrane leaflets[66,67]), or do they actively participate in cholesterol responsivity?

Although the co-evolution with cholesterol-repelling aromatic residues could be coincidental, mutating these residues in presumed functional CRAC motifs, like PD-L1 and TSPO, impairs their cholesterol responsiveness[12,13]. Aromatic residues within these motifs may alternatively facilitate responsivity through the repulsion of cholesterol−with cholesterol acting as a cosolvent for membrane proteins rather

than a ligand[51]—to alter the behavior and functionality of membrane proteins.

Such a sensing mechanism relying on repulsion rather than attraction of the surrounding lipid environment may reflect the membrane saturation sensing mechanism in the transcriptional regulator Mga2, which relies on the relative rotation of two transmembrane domains (TMDs) to sense lipid packing density[68]. Tighter lipid packing favors a rotational orientation, with the bulky tryptophan sensing residue 'hiding' in the dimer interface. Less dense lipid packing in membranes with a high proportion of unsaturated lipid acyl chains favors a different relative orientation of the TMDs, with the sensing residue facing hydrophobic lipid acyl chains, thereby weakening dimer formation.

Analogously, elevated cholesterol levels might induce membrane-exposed aromatic residues and leucines to facilitate dimerization by prioritizing protein-protein interactions over protein-lipid interactions. Dimerization is known to control the functionality of a wide class of both single-pass[62] and multi-pass membrane proteins[69]. Our coarse-grained simulations exploring dimers of the $\gamma$ M4 TMD[23] highlight the role of phenylalanine within the CARC motif in enhancing protein-protein interactions within cholesterol-enriched lipid membranes (Supplementary Fig. 16).

Likewise, elevated cholesterol levels in multi-pass membrane proteins may alternatively force aromatic residues to rotate inward, enhancing interactions with residues in neighboring helices, thereby shielding them from the unfavorable membrane environment. Such an induced structural change could alter protein (channel) configuration and functionality potentially even via long range allosteric coupling[13].

Akin to cholesterol-protein docking studies in a vacuum[3], aromatic residues may however favor cholesterol binding under specific conditions where competition from other lipids is absent. For instance, when these residues are situated within a groove between several transmembrane domains[10,14,32], deeply embedded within the membrane and inaccessible to other lipids except cholesterol, they can effectively promote cholesterol binding. However, it is important to note that this scenario requires knowledge of the protein's full three-dimensional structure, especially for multi-pass membrane proteins. Such comprehensive understanding exceeds the predictive capabilities of models solely based on linear motifs.

The observation of direct binding interactions between aromatic residues within identified CRAC motifs and cholesterol[27] in coarse-grained molecular simulations using the Martini 2 force field appears counterintuitive, given the strong cholesterol repulsion of aromatic residues within this force field. In fact, systematic mutation of aromatic residues within identified CRAC/CARC motifs in native proteins actually increases cholesterol attraction, as described by the same Martini 2 force field (Supplementary Fig. 12). This suggests that secondary interactions, including those with other residues and residues in neighboring helices, as well as the overall three-dimensional protein structure (hydrophobic groves), are likely to play a role in facilitating the observed cholesterol binding.

Despite significant advancements, the mechanisms governing cholesterol-dependent protein regulation in GPCRs remain poorly elucidated. Atomistic simulations revealed that cholesterol binding to specific high-affinity sites reduced $\beta$2AR conformational variability in a high (40%) cholesterol environment compared to a low (10%) cholesterol environment[63]. A primary challenge at elevated cholesterol concentrations lies in distinguishing the effects resulting from cholesterol binding as a weak ligand versus its role as a cosolvent of membrane proteins. Additional control simulations in which cholesterol binding is artificially conserved under low cholesterol conditions, as well as point mutations within the specific binding sites, could further clarify the different roles of cholesterol binding versus its effects on lipid membranes such as stiffening and reduced dynamics.

In summary, our study has demonstrated the ability of Evo-MD to identify evolutionary fingerprints of protein-lipid interactions in membrane proteins. Our methodology relies on the physics-based inverse design of molecules, leveraging the fact that the physical driving forces governing functionality are inherently embedded within the complexity of independently parameterized classical molecular force fields. This approach diverges significantly from prevalent data-driven quantitative structure-activity relationship (QSAR) based inverse design approaches, which employ machine learning based variational encoders to translate optima in an abstract high-dimensional latent space into corresponding chemical structures[35].

By determining the true thermodynamic optimum for cholesterol attraction, Evo-MD has provided insights into the fundamental forces that drive lipid recognition and binding in membrane proteins. This unique ability of Evo-MD enables us to gain a deeper understanding of how proteins recognize and bind specific membrane lipids or lipid-soluble ligands, including hormones and vitamins, within the complex and crowded environment of lipid membranes. We anticipate that physics-based evolution approaches like Evo-MD will unveil insights into the molecular organization of biological membranes and protein trafficking mechanisms[38]. The synergy with other groundbreaking protein structure prediction methodologies, such as the Alphafold 2 project[70], could further facilitate these applications.

## Methods

### Software
Coarse-Grained simulations were performed with the Martini 2.2 and Martini 3 CG force field using the GROMACS 2019.1 molecular dynamics package. EVO-MD is written in Python 3.6.8 and depends on the *NumPy* and *MPI for Python* packages for functionality. Peptide topologies are generated using *seq2itp*[71]. Input parameters for the coarse-grained simulations are based on the Martini 2 'New-RF' parameters[72] and the Martini 3 recommended parameters[43]. with exceptions detailed in the sections below.

### EVO-MD implementation
EVO-MD was developed as a framework for the simulated evolution of MD simulation systems. Simulated evolution is a type of optimization problem involving the optimization of some property of the simulated system, by means of iteratively tuning a set of parameters. The performance (i.e., fitness) of such a parameter set is then measured by means of a fitness function, which generally consists of one or more MD simulations followed by an analysis step.

Using GAs, we can manage large, hyper-dimensional optimization problems through efficient exploration of the search space. Analogous to the method's origin in genetics, we envision each possible solution as a chromosome, which consists of a unique set of parameters encoded into a (bit)string sequence. The algorithm iteratively samples parts of the search space by forming a population of chromosomes and measuring their fitnesses. In line with evolution, individuals with high fitnesses are selected to recombine and form a new population. Since the new population is based on a highest fitness subset of the previous population, it is assumed that the average fitness of the population increases each iteration. This process is visualized in Fig. 1.

Implementation of the cholesterol sensing project is illustrated in Fig. 6. Each candidate peptide is encoded as a sequence of one-letter amino acid codes. For faster convergence, the sequence is mirrored to produce a palindromic sequence, effectively reducing the search space for a peptide 20 amino acids in length from $20^{20}$ to $20^{10}$ (assuming 20 amino acid types). The GA is initialized by generating a random population of $N_{pop}$ sequences, after which each sequence is evaluated in parallel according to the fitness function.

The fitness function takes a sequence as argument and returns a single float value representing the sequence's fitness. This function involves several simulation steps: *generate_peptide*, *insert_peptide*,

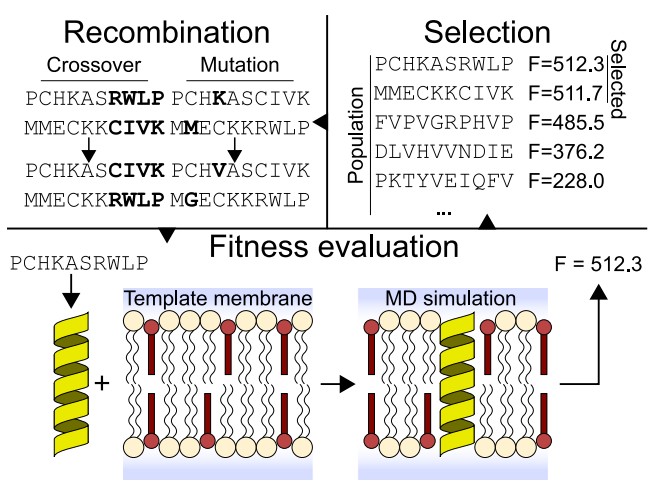

**Fig. 6 | Graphical overview of EVO-MD.** Peptide sequences are evaluated by means of MD simulation. A peptide structure (yellow) is generated from sequence and inserted into a POPC (beige) and cholesterol (red) bilayer membrane. The fitness is then computed from the resulting trajectory. Highest fitness sequences are selected from the evaluated population. Through recombination (involving crossover and mutation operations) of the selected sequences, a new population is generated.

*production*, and *compute_fitness*. *Generate_peptide* generates a peptide structure and topology using the *seq2itp* tool[71], followed by energy minimization and peptide-membrane alignment. *Insert_peptide* combines the peptide structure with an existing equilibrated membrane structure containing 128 lipid molecules (90 POPC, 38 cholesterol) and 1598 Martini water beads, and places the peptide transversely through the membrane. Collisions between peptide and membrane structures are resolved by partially decoupling the non-bonded interactions—combined with soft-core potentials—and running a steepest descent algorithm. The *production* module adds ions to neutralize any net charge on the system, after which equilibration and production simulations are performed. The *compute_fitness* module then measures the ensemble-averaged short-ranged Lennard-Jones interactions between peptide and cholesterol molecules from the simulation trajectory, which is returned as the fitness (Coulomb interactions involving cholesterol are absent within the CG model). Notably, such a fitness is the direct outcome of the competition between cholesterol and POPC lipids to interact with the peptide. Therefore, its value is directly proportional to the adopted cholesterol concentration and thus the relative binding free energy.

Once all sequences in the population have been evaluated, the algorithm proceeds by selecting the best $N$ performers to serve as parents for the next population. A new sequence is generated by recombining two randomly selected sequences from the parent pool, which involves a cross-over operation and a mutation operation. During the cross-over operation, a random position is selected in the new sequence. The part to the left of that position is inherited from the first parent, while the rest of the sequence is inherited from the second parent. Afterwards, the mutation operation ensures that each position in the sequence has a $1/len(sequence)$ chance of being replaced with a random amino acid. New sequences are created in this manner until a new population of size $N_{pop}$ is produced. This process of population fitness evaluation and recombination of the highest fitness candidates into a new population is then repeated until a desired number of iterations is achieved.

A rerun mechanism was implemented to account for possible undersampling during fitness evaluation. If a sequence reoccurs in a future generation, its fitness value will be computed from the weighted average of the current and all prior fitness evaluations. With the chance of sequence reoccurrence increasing as the algorithm converges, this mechanism serves to increase confidence in the final fitness value.

## Membrane setup
The membrane template structure consists of a $5.6 \times 5.6 \times 10$ nm simulation box, containing a bilayer membrane in water solvent. The membrane consists of 90 POPC molecules and 38 cholesterol molecules. The solvent consists of 1598 Martini water beads.

## EVO-MD modules
**generate_peptide.** As the seq2itp tool only produces topology files, a structure file for the peptide is generated by stacking hardcoded amino acid structures along the Z-axis and performing a 1.5 ps simulation at low time step (0.05 fs) using the GROMACS 2019.1 'sd' stochastic dynamics integrator. This allows the hardcoded structure to slowly relax to a more reasonable conformation according to the generated topology.

**insert_peptide.** *Insert_peptide* centers the peptide in the membrane box and merges the two structures together. A steepest descent, combined with a partial decoupling of the non-bonded interactions ($\lambda = 0.75$) and soft-core potentials, is then performed on the merged structure to remove collisions between the peptide and the membrane structures.

**production.** A final steepest descent is performed without soft-core potentials. A short, 1.5 ps simulation is performed at low time step (0.05 fs) using the stochastic dynamics integrator to prevent blowing up of the system before the actual simulation is performed. The production simulation consists of a 500 ns NPT MD simulation with 30 fs time step, of which the first 50 ns are used for equilibration. Temperature is coupled to 300 K using velocity rescaling ($\tau = 1$ ps with separate coupling groups for the membrane, peptide, and solvent), Pressure is coupled semi-isotropically to 1 bar using the Berendsen algorithm ($\tau = 8$ ps), with compressibility set to $4.5 \times 10^{-5}$ bar$^{-1}$.

**compute_fitness.** Evaluation of the sequence's fitness is finalized by computation of a fitness value from the produced simulation trajectory. GROMACS' *gmx energy* tool is used to extract the ensemble average of the non-bonded interaction energies from the production trajectory. The absolute value is then returned to the GA.

Quantification of sequence cholesterol clustering capability was performed by measuring the ratio of cholesterol molecules to membrane molecules within a cylinder of radius $r$ centered on the peptide center-of-mass (COM). GROMACS' *gmx rdf* tool was used to compute a cumulative number radial distribution function ($g_{CN}(r)$) for cholesterol COMs and POPC COMs, both with respect to the peptide COM. The final ratio figures are created by computing:

$$f_{ratio}(r) = \frac{g_{CN,CHOL}(r)}{g_{CN,CHOL}(r) + g_{CN,POPC}(r)} \tag{1}$$

Comparisons between multiple ratio figures (local cholesterol content) were taken at a cylinder radius of 1.0 nm, chosen as a middle-ground between local-sampling (low $r$) and sufficient sampling (high $r$).

## GA parameters
Production runs of the GA were performed according to the parameters as described in Table 1. Parents indicates the size of the selection pool, from which parents were selected at random for the recombination step. Iteration elites describe the number of highest fitness sequences which pass unaltered into the next generation. Rerun elites keeps track of a list of sequences which have been evaluated

**Table 1 | Overview of GA run parameters**

| Population | # of GA runs | Parents | Iteration elites | Rerun elites | Mutation frequency[a] |
|---|---|---|---|---|---|
| 4 | 4 | 2 | 1 | 1 | 1/20 |
| 8 | 4 | 2 | 1 | 1 | 1/20 |
| 16 | 4 | 4 | 1 | 1 | 1/20 |
| 32 | 4 | 8 | 2 | 2 | 1/20 |
| 64 | 2 | 16 | 2 | 2 | 1/20 |
| 128 | 2 | 16 | 2 | 2 | 1/20 |
| 256 | 2 | 16 | 2 | 2 | 1/20 |
| 320 | 1 | 16 | 2 | 2 | 1/20 |

[a]per amino acid.

more than once, and allows several highest fitness sequences to proceed to the next generation unaltered. The total number of elites is equal to the sum of iteration and rerun elites.

**All-atom validation simulations**

Simulations for the analysis of side-chain cholesterol affinity were performed using the GROMACS 2019.1 molecular dynamics package. Simulations for the computation of the free energy profiles and the cholesterol binding residence time were performed using the GROMACS 2021.3 molecular dynamics package, with the Plumed 2.7.2 plugin. Peptides were represented using AMBER99SB-ILDN[73], while POPC and cholesterol were represented with the Slipids forcefield[74,75]. For water molecules we used the TIP3P model. Simulations were performed in the NPT ensemble at 303.15 K, maintained with a Nose-Hoover thermostat. Pressure was kept at 1 bar using a semi-isotropic coupling scheme and a Parrinello-Rahman barostat. Long-range electrostatic interactions were calculated using the PME algorithm with a real-space cutoff of 1.4 nm. Van der Waals interactions were calculated with a 1.4 nm cutoff, and dispersion corrections for energy and pressure were applied. The leap-frog algorithm with a time step of 2 fs was used to integrate the equations of motion. The LINCS algorithm was used to constrain hydrogen atom-containing bonds.

**Analysis of side-chain cholesterol affinity.** Lipid bilayer simulation systems were set up consisting of 83 POPC lipids, 35 cholesterol molecules, and 6359 water molecules. Peptides were generated in an initial helical conformation and placed transversely through the membrane, no bias was enforced during the simulations. The systems were equilibrated for 50 ns with the lipids and peptide coupled to a 600 K temperature bath, while water remained at 303.15 K. After initial equilibration, a simulated annealing procedure linearly decreased the temperature of the lipids and peptide from 600 K to 303.15 K over 10 ns, after which the simulations continued for another 50 ns at 303.15 K. 2 μs measurement simulations were performed for each sequence, of which the first 250 ns were discarded for equilibration purposes. 5 replicates were performed for each sequence according to this procedure. The error bars represent the maximum difference among the five ensemble averages from these five simulations.

**Computation of the free energy profiles.** Umbrella sampling (US) was used to determine the free energy profile of cholesterol binding to KALP21, L11, γ M4, and the mutant of γ M4. As the reaction coordinate, we used the in-plane center-of-mass distance (xy-distance) between the cholesterol ring system and all the peptide Cα atoms located in the same membrane leaflet as the cholesterol molecule (residues 1–11 and 1–12 for KALP21 and L11, respectively; residues 1–14 were selected for the γ M4 transmembrane peptide and its mutant). To describe the binding process, we sampled the 0.7–2.3 nm range of xy-distance using 9 evenly spaced US windows separated by 0.2 nm. In each

window the reaction coordinate was subject to a harmonic bias potential with a spring constant of 250 kJ mol$^{-1}$ nm$^{-2}$. For each window, 1.5 μs simulations were performed, and the free energy profiles were calculated using the WHAM method. For each window, the first 400 ns of the trajectory were discarded for equilibration purposes. For each peptide we simulated three replicas, each starting from an independent set of configurations, to produce the final PMFs. The statistical uncertainties of the free energy were estimated using the Monte Carlo bootstrap method, taking into account autocorrelation times.

**Residence time of cholesterol binding.** The residence time of cholesterol binding to the L11 peptide was calculated according to the following formula (see Zwanzig[76]):

$$\tau(a \rightarrow b) = \int_a^b dx \frac{e^{\beta G(x)}}{D(x)} \int_{x_0}^x dy e^{-\beta G(y)} \qquad (2)$$

where $x$ is the reaction coordinate (i.e., L11–cholesterol xy-distance), while the integration limits $a$ and $b$ correspond to the bound and dissociated states, respectively (i.e., 0.70 and 1.80 nm). $G(x)$ and $D(x)$ represent the free energy and diffusion coefficient as a function of the reaction coordinate $x$. $x_0$ represents the position of a reflecting barrier at 0.62 nm.

To obtain $D(x)$, the diffusion coefficient was computed for each US window separately according to $D = \text{Var}(x)/\theta$[77] and interpolated. Here, $\text{Var}(x)$ and $\theta$ represent the variance and autocorrelation of the reaction coordinate in a given US window.

**Restraining vertical lipid mobility/flip-flopping**

To investigate the hydrophobic mismatch mechanism, the removal of vertical mobility of lipids and lipid flip-flopping was facilitated by applying an inverse flat bottom position restraint to the first beads of POPC (NC3 bead) and cholesterol (ROH bead). The position restraint consists of a layer, parallel to the membrane and centered on the bilayer center. A harmonic force with force constant 1000 kJ.mol$^{-1}$.nm$^{-2}$, directed away from the bilayer center, is applied to affected beads that come within 2.0 nm (NC3) or 1.5 nm (ROH) of the center of the bilayer.

**Database analysis using a convolutional neural network**

**Convolutional neural network.** The CNN architecture consisted of a one-hot encoding step, which is fed into 2 convolutional layers (128 nodes each) with max pooling, followed by 2 fully-connected dense layers (36 nodes each) and a single output neuron. The random dropout, which is applied before the output of the convolutional layers enters the dense layers, was set to 0.5%. A dataset of 26769 sequences generated using Evo-MD was used for the development of the CNN model, of which 20% was used as an independent validation set for the final model. The remaining 80% of the dataset was used in a 4-fold cross-validation (each fold using 5353 sequences as a test set, and 16061 sequences for training). The model was trained in 16 epochs, with a batch size of 64 and a learning rate of 0.001. An independent benchmarking of the model's performance against molecular dynamics simulations over the whole applicability domain (Coefficient of determination: $R^2 = 0.859$) is given in Supplementary Fig. 18.

**Database analysis.** Protein sequences and corresponding transmembrane predictions were downloaded from the TmAlphaFold Transmembrane Protein Structure Database (https://tmalphafold.ttk.hu/downloads). From this database, *Homo sapiens* (UP000005640), *Mus musculus* (UP000000589), and *Rattus norvegicus* (UP000002494) were considered for analysis. We only included proteins that passed all 10 TM prediction quality flags (i.e., categorized as 'excellent'), as described in ref. 78. The resulting dataset contained 8370 protein entries in total, which was subsequently split in a single-pass dataset (2084 entries) and a multi-pass dataset (6286 entries, 42436 passes).

We post-processed these datasets to produce sequences of 20 amino acids, as the CNN was trained on this type of data. TM sequences that exceeded 20 amino acids in length were removed, and TM sequences shorter than 20 amino acids were extended evenly along the edges using the corresponding non-TM amino acids from the protein sequence. We ended up with 902 single-pass sequences and 11,954 multi-pass sequences, which we used for fitness prediction using the CNN, and subsequent analysis.

### NMR and CD analysis

**Sample preparation.** Membranes were prepared using standard protocols for the hydration of lipid films[79]. Briefly, 5 mg of 1,2-di-O-dodecyl-sn-glyercero-3-phosphocholine (12:0 ether-linked DLPC lipid), 0.2 mg of labeled peptide (labeled at the carbonyl of the two leucine-proximal lysine residues) and 1.093 mg of cholesterol ($^{13}$C labeled at C4) were dissolved in chloroform. The chloroform was then dried with gentle $N_2$ flow and the film was stored under vacuum overnight for complete evaporation of chloroform. The film was then hydrated with 250 μL of buffer (mixture of 5 mM HEPES buffer, pH 7.4 and 100 mM NaCl). The hydrated film was then sonicated (5 min on, 10 min off, 4 cycles in a 25 °C water bath) to prepare the final membranes. The sample was lyophilized and thoroughly mixed with a solution of $^{13}$C-depleted $d_8$-Glycerol (60 percent by volume), and 0.13 mg of AmuPoL. A sample without $^{13}$C labeling of the peptide provided a control.

**Circular dichroism measurements.** CD spectra were recorded in a Jasco J815 spectrometer with a scan rate of 20 nm/min. For the CD measurement, liposomes were prepared in the same way as for the NMR sample, but with 2 mM phospholipids and 0.5 mM cholesterol. The phospholipid composition was an equimolar mixture of 1,2-ditetradecanoyl-sn-glycero-3-phosphocholine (DMPG) and 1 mM 1,2-Dimyristoyl-sn-glycero-3-phospho-rac-(1-glycerol) (DMPG). The peptide concentration was 100 μM.

**DNP enhanced ssNMR measurements.** All DNP-enhanced NMR spectra were recorded with a 600 MHz Bruker Avance III HD spectrometer (magnetic field of 14.1 T) equipped with 3.2 mm low temperature (LT) HCN magic angle spinning (MAS) DNP probe. A 395 GHz gyrotron oscillator was deployed to deliver the desired microwave irradiation to the sample through a corrugated waveguide. For the LT MAS probe, variable temperature, bearing and drive gasses were cooled with a second-generation Bruker liquid nitrogen cold cabinet, operating at 100 K. Samples were packed into 3.2 mm zirconia MAS NMR rotors via a custom-made filling device made from a truncated pipette tip. Finally, the rotor was centrifuged to ensure proper packing. $^{13}$C Proton-driven spin diffusion (PDSD) spectra[80,81] were carried out at 8 KHz MAS. Cross polarization from proton to carbon was implemented with a 1.5 ms Hartmann-Hahn transfer using 66–74 kHz (10% linear ramp) on the proton channel, and 71 kHz on the $^{13}$C channel. Decoupling, 83 kHz SPINAL-64[82], was applied on the proton channel during acquisition. A PDSD mixing time of 30 s was chosen to effect transfer over the expected distance range of about 6–9 Å. Spectra were referenced by setting the $^{13}$C signal from silicone to 4.3 ppm on the DSS scale[83]. All spectra were acquired and analyzed in Topspin 3.5 patch level 6.

### In-vitro expression of short hydrophobic sequences

**Cloning.** Amino acid sequences for $D_3K_3L_{10}K_3D_3$ (L10), $D_3K_3L_{11}K_3D_3$ (L11), and $GK_2[LA]_7LK_2A$ (KALP21) were introduced into mScarlet-N1 and mEmerald-N1 by Gibson assembly. The final constructs were all confirmed by sequencing (Supplementary Table 1).

**Cell-based experiments.** HEK cells were transfected by Lipofectamine 2000 (Thermo Fisher Scientific) following manufacturer's instructions.

Briefly, 3 ml of Lipofectamine 2000 was mixed with max 2 mg of total DNA (in equimolar ratio) in 200 ml OptiMEM (Gibco). Transfection mix was incubated for 30 min at room temperature, then was added to the cells. Cells were transfected and incubated overnight (37 °C and 5% $CO_2$). The day after medium was fully replaced with fresh supplemented DMEM.

Prior to imaging the transfected cells were incubated with Wheat Germ Agglutinin, CF®405S Conjugate (WGA405, 1:20, stock 2 mg/ml) for 10 min. During acquisition the laser power was kept constant. Exposure time 200 ms and Piezo stage z-motor was used to collect z-stacks.

For visualizing the intracellular organelles, the transfected cells were incubated with LysoTracker™ Deep Red, (Thermo Fisher Scientific) and MitoTracker™ Deep Red FM, (Thermo Fisher Scientific) for visualization of lysosomes and mitochondria, respectively. Fluorescent dyes were diluted 1:1000 in pre-warmed imaging solution and added to HEK cells 10 min before imaging.

**Image analysis.** Images were acquired using Acquisition software NIS Elements 5.21.02 and analyzed with ImageJ (NIH). Freehand selection tool in Fiji was used to select a region of interest (ROI) of 3 pixel width following the fluorescence signal of WGA405 (i.e., 405-channel) as reference for plasma membrane from the medial cell plane. Each ROI was assessed for the signal intensity in the 561-channel (for mCherry-CD36) and the mean fluorescent intensity was measured from the ROI for calculating Intensity/length (mm). GraphPad Prism 9 was used to plot the graphs (each value is shown as the average ± standard error of the mean). Statistical test performed was unpaired t-Test ($p < 0.0001$, $P$ value summary ****).

### Reporting summary

Further information on research design is available in the Nature Portfolio Reporting Summary linked to this article.

### Data availability

The datasets generated by Evo-MD used in this work are available at [https://doi.org/10.5281/zenodo.15925656]. The trained CNN model used in this work is available at [https://doi.org/10.5281/zenodo.15925656]. All relevant data supporting the findings of this study are available with the paper and its supplementary information files. Source data is provided with this paper.

### Code availability

The version of Evo-MD used in this work is available at [https://doi.org/10.5281/zenodo.15925656].

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

## Acknowledgements

The Dutch Research Organization NWO (Snellius@Surfsara) and the HLRN Göttingen/Berlin are acknowledged for provided computational resources. J.M. and H.J.R. also gratefully acknowledge the Gauss Centre for Supercomputing e.V. (www.gauss-centre.eu) for funding this project by providing computing time through the John von Neumann Institute for Computing (NIC) on the GCS Supercomputer JUWELS at Jülich Supercomputing Centre (JSC), and on the HAWK supercomputer at the High-Performance Computing Center Stuttgart (HLRS). We thank Advanced Medical Bioimaging Core Facility at Charité, Berlin, for the support. D.M. is supported by the start-up funds from DZNE, the grants from the German Research Foundation (MI 2104 and SFB1286/B10) and the ERC Grant MemLessInterface (101078172). P.C. and J.C. gratefully acknowledge financial support from the National Science Centre, Poland (grant no. UMO-2021/41/N/ST4/03571). P.C. and J.C. also gratefully acknowledge Polish high-performance computing infrastructure PLGrid (HPC Center: ACK Cyfronet AGH) for providing computer facilities and support within computational grant no. PLG/2023/016277. J.M. and H.J.R. thank the NWO Vidi scheme (project number 723.016.005) for funding. J.M. was additionally funded by the Deutsche Forschungsgemeinschaft (DFG, German Research Foundation) under grant number RI 2791/7-1. H.J.R. was additionally funded by the Deutsche Forschungsgemeinschaft (DFG, German Research Foundation) under Germany's Excellence Strategy-EXC 2033-390677874-RESOLV.

## Author contributions

J.M. and H.J.R. designed the research. J.M. developed the Evo-MD code and performed CG MD simulations. N.V., N.v.H,, and J.M. developed the neural network code and performed the database analysis. C.H., R.S., H.W., and D.M. performed and analyzed the in-vitro cell experiments. P.C. and J.C. performed and analyzed all-atom MD validation simulations. P.P. and L.A. performed and analyzed the NMR and CD experiments. D.A. and A.K. synthesized peptides for the cell, NMR, and CD experiments. J.M. and H.J.R. wrote the manuscript.

## Funding

## Competing interests

The authors declare no competing interests.
