## [Transparent Peer Review file · Nature Communications]

Physics-Based Evolution of Cholesterol-Attracting Transmembrane Helices: Deciphering Cholesterol Attraction in Native Membrane Proteins

Corresponding Author: Professor Herre Risselada

Version 0:

Reviewer comments:

Reviewer #1

(Remarks to the Author)

This work introduces an innovative approach of Evo-MD to uncover the cholesterol-recognition motifs. The results are presented clearly and effectively, making it easy to understand the findings. I would recommend for the publication after addressing following points:

1. The overwhelming advantages of Evo-MD should be discussed more.
2. What is the specific reason to select 30% cholesterol and 70% POPC membrane?
3. Is it a complete manuscript? Why the supporting figure is starting from Figure S7, and the supporting table is starting from Table S2?
4. All the peptides tested in this manuscript should be summarized in a table and numbered.

(Remarks on code availability)

Reviewer #2

(Remarks to the Author)

Review attached.

(Remarks on code availability)

Reviewer #3

(Remarks to the Author)

(Remarks on code availability)

Version 1:

Reviewer comments:

Reviewer #1

(Remarks to the Author)

Thanks for addressing my concerns. I am ok to move forward.

(Remarks on code availability)

Reviewer #2

(Remarks to the Author)

I am satisfied with the answers to my review and the revised manuscript. Overall, I can recommend publication. However, I disagree with the authors on the matter of convergence of the PMFs. For example, in the case of yM4 two replicas show a weak binding minimum, but replica 2 is repulsive in the same region. KALP21 also has one different profile, and the binding minima in the case of yM4 mutant are also different in all three PMFs. The authors should at least acknowledge this fact in the manuscript. As far as I can tell, the SI PMF profiles are also not referenced in the main paper, even though I might have missed it due to the lack of track changes. The authors should also add appropriate data and code availability statements.

(Remarks on code availability)

Code was not made available for review.

Reviewer #3

(Remarks to the Author)

(Remarks on code availability)

Reviewer #4

(Remarks to the Author)

In this study, the authors developed a computational method, termed Evo-MD, and applied it to investigate cholesterol binding to peptides of a fixed-length sequence of 20 amino acids within a transmembrane domain. The study is overall very systematic and detailed and quite well performed. This includes also in particular the solid-state NMR data, however not the way how those are described in the text. If the authors can fix the points below I would support the publication of this study in Nature Communications.

Major points:

- Doesn't cholesterol bind also at the interface between two transmembrane helices? Would it not be important to study helical hairpins rather than single TM helices?

- "Such binding mode is confirmed both by our molecular dynamics simulation as well as solid state NMR experiments (see Fig. 4A).":

There is no description at all of the NMR results in the main text. As the manuscript overall is quite lengthy and extremely detailed it is surprising that the NMR results are neither described nor discussed. Not with a single sentence?!

Especially given that solid-state NMR data are used as a selling point in the abstract: "Additional support for these findings is provided by atomistic simulations and solid-state NMR experiments"

- Figure 4: "Interaction between cholesterol (C-4) and the deeply located lysine residues (carbonyl of positions 6 and 15 in the sequence logo) is also observed in DNP-enhanced ssNMR (inset).":

The inset is too small. A larger spectral excerpt needs to be shown – not just in the SI. The way the single peak is shown without the rest of the spectrum is suboptimal – and extremely unusual in the field. A larger spectral section allows to assess spectral resolution, S/N, line splittings etc. much better.

Minor points:

- Experimental section: Please provide details about decoupling in ssNMR experiments, etc.

- „based on a reinforcement value measured during the simulation (See ref. ?? for a recent perspective on EVO-MD)“ -> Please check carefully all refs

also: How can there be a recent perspective on Evo-MD, which is described for the first time in this manuscript??

- Fig. S11. The PDS mixing time was 15 seconds. ??? Normally I would expect a mixing time in the ms range.

- Same for Fig. S12

(Remarks on code availability)

Version 2:

Reviewer comments:

Reviewer #4

(Remarks to the Author)

The authors have improved the solid-state NMR aspects of the manuscript significantly and I support now acceptance of the manuscript in Nature Communications.

(Remarks on code availability)

In the following sections, we meticulously address each point raised by the reviewers.

Reviewer 1:

Reviewer "1. The overwhelming advantages of Evo-MD should be discussed more.":

Author reply: We thank the reviewer for realizing the unique and significant advantages of Evo-MD. The main advances of Evo-MD are in unraveling the molecular mechanism of membrane-related processes such as the sensing of distinct lipid compositions, membrane curvature, and membrane phase (e.g., liquid ordered vs liquid disordered) as well as the design of novel sequences for biomedical application purposes (e.g., membrane-targeting antiviral sequences). We have recently written a special perspective on the application of physics-based design methods that also highlights promising preliminary applications of the Evo-MD method, with the current study on cholesterol being the central example of such an application. Notably, the current work on cholesterol binding is the original and first paper presenting the Evo-MD method. Instead of publishing the method alone in a computational journal, we chose to demonstrate the method through a direct research question based application paper with combined experiments. Obviously, the time scale required for such an extensive combined work (years) far exceeds that of a standard method paper. The revised version of the paper now also serves as a reference/benchmark for commonly used force fields in the study of cholesterol binding, as we were in a unique position to systematically address this issue (upon Martini community request) and wanted to further emphasize the consistency of our conclusions. In addition, we added some emphasis on the potential and application of Evo-MD in the conclusion section and also refer to our very recent perspective on physics-based design methods.

Reviewer "2. What is the specific reason to select 30% cholesterol and 70% POPC membrane?":

Author reply: 1) Various cholesterol ratios occur, depending on cell and membrane type. The plasma membrane of common cell types (leukocytes, epithelial cells, neurons, and mesenchymal cells; ref. 1) consists of about 30 mol% cholesterol. In addition the majority of the lipids consists of monounsaturated PC lipids (one double bond in a single tail), which is why POPC is a stereotypical model lipid. Although cholesterol/PC ratio affects properties of the bilayer, which would be interesting to study systematically, the resource-intensive nature of the method complicates this.

2) Sampling statistics improve the closer the cholesterol/POPC ratio is to 1 due to increased observation of cholesterol—peptide (and POPC—peptide) interactions. However, phase-separation can start to occur at higher cholesterol/PC ratios. ([https://doi.org/10.1016/S0009-3084\(98\)00006-1](https://doi.org/10.1016/S0009-3084(98)00006-1), <https://doi.org/10.1016/j.bbamem.2016.08.005>)

3) Finally, we emphasize that good cholesterol attractors in POPC:cholesterol membranes are also good attractors in complex plasma membrane mimics (Fig. S8) as well as DPPC:cholesterol membranes in a more ordered phase. Therefore, the thermodynamic principles underlying the mechanism of cholesterol attraction are rather independent of the membrane model.

Reviewer "3. Is it a complete manuscript? Why the supporting figure is starting from Figure S7, and the supporting table is starting from Table S2?":

Author reply: Figures 1–6 indicate the main text, therefore the supplementary section continues with S7 and so on. Table 1 is in main text, next table in SI is S2. The numbering is somewhat subject to taste.

Reviewer "4.All the peptides tested in this manuscript should be summarized in a table and numbered.":

Author reply: The peptides are listed in Table Fig. S23. This plots the MD-calculated fitness (total protein-cholesterol potential energy) against the binding potential energy of an identified bound single cholesterol molecule. If there are several binding event simultaneously, we pick the strongest binder based on the

measured average contribution to the potential energy (Max. single cholesterol interaction)). However, the main purpose of the combined plot and table Fig. S23 is to illustrate the strong correlation between the total protein-cholesterol potential energy and the binding affinity for single cholesterol molecules.

We now better refer to this provided list of peptides in the main text.

Reviewer #2 & #3:

Reviewer "1. The system sizes for the coarse-grained simulations are surprisingly small and significantly smaller than the usual membrane sizes considered even for parameterization purposes. Therefore, we consider it important to verify how much the fitness score changes with system size. Investigating how the fitness score changes with progressively larger systems for at least several protein sequences generated by the GA seems easy enough of a test.":

Author reply: We thank the reviewer for pointing this out. We have now performed additional simulations to confirm the independence of membrane size on fitness (Figure S25). A potential effect of membrane size on fitness values is more likely to occur in the extreme hydrophobic mismatch regime due to the intermediate nature of the membrane thickness adjustment (several nanometers). However, a system-dependent change in fitness value does not imply that the optimum in sequence space—which is the main goal of our study—will also change. In fact, we find that the optimum in sequence space is rather independent of membrane composition, even though the measured fitness is highly dependent on membrane composition. For example, a peptide optimized in a simple POPC:cholesterol system also performs well in a complex membrane model that mimics the plasma membrane (see Fig. S8). In other words, the principles underlying cholesterol attraction are most likely rather invariant to membrane composition as well as membrane size effects. Intriguingly, in the revised manuscript, we have now performed all Evo-MD simulations with both the Martini 2 and the newer Martini 3 force-field. These additional simulations

underscore that, even though the measured fitness and the associated cholesterol attraction may differ significantly between different force-field versions, the thermodynamically determined design rules of cholesterol attraction (e.g., hydrophobic mismatch combined with a preference for small hydrophobic amino acids) are invariant to the force field version.

Reviewer "2. In our view, the link between Cholesterol attraction and the size of the amino acid is insufficiently established. It appears to be true that there is a correlation between fitness and some smaller amino acids (e.g. PRO, VAL). However, the bond scaling analysis seems unnecessarily artificial. It also seems suspicious that the smallest hydrophobic amino acid Alanine is not present to a significant extent. Instead, it would make more sense to correlate the fitness with the SASA or SASA inferred volume of the hydrophobic domain. Ideally, this analysis could be done on the all-atom level after backmapping. If it is truly just the slenderness there should be a pretty good correlation with the SASA.":

Author reply: We fully agree with the reviewer that the lack of attraction for alanine remained somewhat paradoxical and unsatisfactory. To this end, we replaced the simulations in which we scaled the bond length of valine with actual simulations of all relevant hydrophobic amino acids of different sizes (Fig. 3C) for three different force fields (Martini 2, Martini 3, all atom (AMBER99SB-ILDN with Slipids)). In addition, we reproduced *all* of the Evo-MD simulations using the newer Martini 3 force field. This part of the revision was a considerable amount of work (way over the 4 weeks revision time policy). However, the results obtained are very informative.

In fact, it turns out that the conclusion—relative cholesterol affinity prefers smaller hydrophobic amino acids—is correct, but that alanine is simply not well represented in the Martini 2 model. The small side chain bead that the Martini 2 model uses to model P, V and L over-represents cholesterol attraction. Alanine does not contain such a side chain bead in Martini 2, which is why it shows lesser

affinity in M2. In Martini 3, the methyl side chain in alanine is modeled by a very small dedicated bead type (1:1 mapping). Consequently, glycine and alanine show the highest relative (!) affinity for cholesterol in both AA and M3 simulations. Moreover, all three force fields agree on the repulsive effect of large aromatic hydrophobic amino acids (particularly in case of Phenylalanine—a key residue within the CARC motif). The ranking of hydrophobic amino acids on size (X-axis in Fig. 3C) illustrates that relative cholesterol affinity in atomistic simulations is indeed largely amino acid size determined, and seems surprisingly independent on amino acid hydrophobicity. We propose that bulky, highly corrugated proteins more disrupt the order within the surrounding cholesterol matrix, resulting in a local depletion of cholesterol. The absence of correlation between amino acid hydrophobicity and (relative) cholesterol affinity suggests that such an effect is likely driven by optimizing cholesterol-cholesterol interactions rather than protein-cholesterol interactions. Hydrophobic transmembrane domains therefore tend to show a net repulsion rather than an net attraction towards cholesterol. This repulsion is however compensated by negative hydrophobic mismatch via lysines and arginines exposed to the hydrophobic membrane core.

Reviewer "3. Can the authors comment on why V, and P, occur at positions 1, and 2 of the helix sequence according to Figure 3A? This part of the peptide should be in the hydrophilic / head-group region of the membrane. Therefore, it is strange to see such a significant population in that position.":

Author reply: Sequences are optimized for maximum fitness (i.e. cholesterol attraction). Residues at specific positions are selected accordingly. To maximize cholesterol attraction, the peptide must be transmembrane, have a hydrophobic mismatch, and contain optimal cholesterol-attracting residues at positions where cholesterol interacts with the peptide. Residues at positions 1 and 2 do not interact directly with cholesterol, which is deeper in the membrane, in part because of the hydrophobic mismatch caused by residues 3–6. We expect that positions 1 and 2 play only a minor role in altering the transmembrane

orientation of the peptide, since induction of a small tilt may increase exposure to the hydrophobic region of the membrane, thereby increasing cholesterol exposure. The evolutionary pressure on these positions is therefore very low compared to the other positions, resulting in low convergence and (mostly) random residues. Note also that this figure combines many sequences into one sequence logo, without taking into account the correlation between specific positions. Although hydrophobic residues can occur at these positions, it is unlikely that both positions will be hydrophobic in the same peptide, as this has a higher chance of destabilizing the transmembrane position.

Reviewer "4. From the analysis of predicted fitness scores for the TmAlphaFold database, the authors conclude that hydrophobic mismatch plays a minor role in real biological systems and instead cholesterol attraction must be driven by hydrophobic slenderness. However, no actual proof is offered at this stage. We suggest correlating the fitness score with the occurrence of smaller hydrophobic amino acids and/or the SASA of the hydrophobic domain.":

Author reply: We now address this point with the extensive additional atomistic simulations provided in Figure 3C, which systematically rank the hydrophobic amino acids along the x-axis with increasing size, while clearly showing a decreasing trend in relative affinity for cholesterol.

Furthermore, the illustration that strong hydrophobic mismatch plays a lesser role in real biological systems is now put more on the foreground via the live cell experiments which were manipulated to express the labeled trans-membrane domains L10 and L11. These are quite exciting experiments that deserve more credits than the SI. Though the regime of strong net attraction mediated by mismatch is likely not reached in nature, we emphasize that mismatch enhances relative cholesterol affinity (reduces cholesterol repulsion) all the way to the biologically common hydrophobic length of 20 hydrophobic amino acids.

Finally, in the revised manuscript, we prefer to put lesser emphasis on mismatch claims derived from scanning the TmAlphaFold database with the EvoMD trained CNN prediction models, as both Martini models can be somewhat inconsistent for some residues when compared to atomistic simulations, and these effects may somewhat confound the true effect of mismatch. For example, the presence of leucine and valine will strongly enhance cholesterol attraction based on M2 predictions, while M3 predictions will underestimate the repulsion of aromatic residues compared to atomistic simulations.

Reviewer "5. One key piece of evidence is the all-atom PMF presented in Figure 3. There is a growing consensus in the MD community that PMFs generated by umbrella sampling (US) are very susceptible to how initial starting configurations are generated, the window spacing, and the force constant. A recent study by Aho et al. suggests even for well- defined 1D reaction coordinates there can be as much as 2-20 kcal/mol in difference. In light of these results, it is surprising that the presented PMFs appear to be only one replicate. Usually, a single replicate is insufficient and not in line with the best practices of the field. We suggest adding at least two replicates and providing the histograms in the SI so that the sufficient overlap criterion can be verified.":

Author reply: We appreciate the reviewer's insightful comment. The shape of the PMF obtained through umbrella sampling (US) can indeed be highly dependent on the initial configurations of the system.

In response to the reviewer's suggestion, for each peptide under investigation, we prepared three independent replicas (with trajectory lengths of 1.5 μ s per window), each starting from a different set of initial configurations to generate the final PMFs. In total, we obtained trajectories totaling 40 μ s per system. Since no significant energy barriers were observed in our system, this should ensure adequate convergence of the profiles. For comparison, the study cited by the reviewer, which investigated the binding of the LLL tripeptide to clpS protease,

achieved a total trajectory length of 10.5 μ s per system, while other systems in that study had significantly shorter simulation times.

For each replica, we analyzed the histograms to confirm that the reaction coordinate was sufficiently sampled throughout the full range (see Fig. S28). Additionally, we plotted the PMF curves for each replica individually (Fig. S27), revealing only minor variations within each set. The final PMF (shown in Fig. 3), combining data from all three replicas, is consistent with those presented in the original manuscript. Moreover, the reduction in PMF statistical uncertainty significantly enhanced the reliability of our conclusions regarding the differences in cholesterol affinity for each peptide.

Furthermore, in the revised manuscript we now put the extremely short first passage time (retention time) of cholesterol unbinding from linear motifs (even at the limit of thermodynamic optimality) measured via free energy calculations into more contextual terms, as this would limit the biological utility of such a mechanism. For example, other ligands known to alter GPCR function bind with retention times exceeding millisecond time scales, whereas cholesterol binding to isolated linear motifs—even when close to optimal—is highly dynamic with sub-microsecond retention times.

Reviewer "6. The description of the training procedure for the NN is incomplete. The following question should be answered in the SI or method section: What exactly were the CNN inputs? How big were the training and testing data sets? How does CNN perform on both?":

Author reply: We have added the missing information, as well as a benchmark of the model's performance against independent molecular dynamics simulations.

Reviewer "7. Usage of term 'sensing': Sensing requires recognition and response. As we do not see evidence of any response, we suggest avoiding this term.":

Author reply: We agree with the reviewer's point. Our focus is on cholesterol attraction and binding. We have rephrased this term throughout the manuscript.

Reviewer "8. Some of the figures and captions need to be improved: a. Fig. S3: The gray background of panels D,F,E,G is confusing. Does it have any meaning?":

Author reply: The Gray background emphasized the atomistic nature of these figures, in contrast to the coarse-grained figures A and B. We have removed it to improve clarity.

Reviewer "b. Fig. S3: broken caption: "non-CARC (F- ζ A)""":

Author reply: Corrected.

Reviewer "c. Fig. S8: What are the colors? What does this plot show (averaged over 2me or a single snapshot?) How is the % calculated?":

Author reply: The colors represent the different lipids of the native epithelial membrane model, in contrast to the simple POPC/cholesterol systems used during the evolution. The % shows the ratio of cholesterol to all membrane lipids within a radius from the peptide, computed in the same manner as in the main text figures (see method section "Module: compute_fitness").

Reviewer "d. Fig. S15: What are the black below molecules?":

Author reply: Cholesterol. The caption has been updated to reflect this.

Reviewer "e. Fig. S16-S19: The contrast of the especially yellow lines is very poor and therefore hardly visible.":

Author reply: A different color and outline has been picked to improve the clarity of the figure.

Reviewer "f. p16/17: References broken to Fig. 5":

Author reply: Fixed

Reviewer "9. From the current manuscript, it is not obvious if and where the code and data are available. We suggest providing the evoMD code as well as the NN code online. In addition, providing the results of the evoMD at least in terms of sequences and fitness score seems appropriate.":

Author reply: Code and data will be uploaded on Github or some public platform according to journal/publisher recommendations.

Prof. Dr. Herre Jelger Risselada

Department of Physics TUDortmund
| Otto-Hahn-Str. 4 | 44227 Dortmund

Prof. Dr. Herre Jelger Risselada, Department of Physics TUDortmund, Otto-
Hahn-Str. 4, 44227 Dortmund

Springer Nature Group

Dortmund, April 3, 2025

Dear Editor,

I am writing to submit the revised version of our manuscript titled "*Physics-Based Evolution of Cholesterol-Attracting Transmembrane Helices: Deciphering Cholesterol Attraction in Native Membrane Proteins*". We sincerely thank the reviewers and editor again for their highly positive and constructive comments and the time invested in evaluating our work.

We have carefully considered all the remaining suggestions received. We have thoroughly anticipated on the reviewers' 4 suggestion to better emphasize on the performed NMR experiments within the main manuscript and provide all detailed information and data on these experiments.

We appreciate the opportunity to address the reviewers' comments and look forward to the editor's decision on our revised submission.

Yours sincerely,

Jeroen Methorst & Herre Jelger Risselada

✉ jelger.risselada@tu-dortmund.de

In the following sections, we meticulously address each remaining point raised by the reviewers.

Reviewer #2 (Remarks to the Author):

Reviewer "I am satisfied with the answers to my review and the revised manuscript. Overall, I can recommend publication. However, I disagree with the authors on the matter of convergence of the PMFs. For example, in the case of yM4 two replicas show a weak binding minimum, but replica 2 is repulsive in the same region. KALP21 also has one different profile, and the binding minima in the case of yM4 mutant are also different in all three PMFs. The authors should at least acknowledge this fact in the manuscript. As far as I can tell, the SI PMF profiles are also not referenced in the main paper, even though I might have missed it due to the lack of track changes. The authors should also add appropriate data and code availability statements.":

Author reply: The SI PMF plots are now explicitly being referred to in the main manuscript.

We note that the key insight regarding PMF convergence lies in the significant differences between the obtained PMF ensembles across distinct sequences (e.g., L11 vs KALP21), rather than the consistency of multiple PMFs of the same sequence. Nevertheless, we agree with the reviewer that obtaining as well as proven full convergence of atomistic PMF remains a delicate issue.

Reviewer #2 (Remarks on code availability):

Code was not made available for review. We provided the Evo-MD code with the resubmission of the revised manuscript.

Reviewer 4:

Reviewer "Doesn't cholesterol bind also at the interface between two transmembrane helices? Would it not be important to study helical hairpins rather than single TM helices?":

Author reply: Our hypothesis addresses the potential existence of cholesterol-binding linear motifs within transmembrane domains. Consistent with CRAC/CARC motifs, which envision cholesterol docking with isolated transmembrane helices derived from membrane proteins (10.3389/fphys.2013.00031), we investigated how isolated membrane helices facilitate cholesterol binding. As suggested by the Reviewer, cooperation between neighboring helices (helical hairpins), particularly through hydrophobic grooves, enhances cholesterol-binding potency compared to the situation where cholesterol binding would be driven by individual helices as being envisioned by the concept of linear binding motifs. This finding aligns with our manuscript's main conclusions.

We now state our research hypothesis in the introduction of the main manuscript as, "However, the looseness of the CRAC and CARC definitions, represented via the flexible algorithmic rules: (L/V)-X₁₋₅-(Y)-X₁₋₅-(K/R) and (K/R)-X₁₋₅-(Y/F)-X₁₋₅-(L/V) respectively, is rather unexpected for a motif that mediates binding to a unique molecule, raising skepticism about its predictive value []. Therefore, this flexible definition based solely on residue patterning within a single transmembrane motif neglects the overall 3-dimensional protein structure of multi-pass membrane proteins, such as the presence of hydrophobic grooves and cavities formed between helical hairpins and additional adjacent transmembrane helices, which have been shown to actively mediate cholesterol binding."

Reviewer "- "Such binding mode is confirmed both by our molecular dynamics simulation as well as solid state NMR experiments (see Fig. 4A).": There is no description at all of the NMR results in the main text. As the manuscript overall is quite lengthy and extremely detailed it is surprising that the NMR results are neither described nor discussed. Not with a single

sentence?! Especially given that solid-state NMR data are used as a selling point in the abstract: “Additional support for these findings is provided by atomistic simulations and solid-state NMR experiments”:

Author reply: We have extended the discussion on the performed NMR experiments within a dedicated result section. In addition we have extended on the NMR results shown in Figure 4 within the main manuscript. Additionally, we have added directional arrows to highlight the isotope-labeled carbon atoms in the molecular simulation snapshots showing cholesterol binding to deep lysines.

Reviewer "Figure 4: “Interaction between cholesterol (C-4) and the deeply located lysine residues (carbonyl of positions 6 and 15 in the sequence logo) is also observed in DNP-enhanced ssNMR (inset).”: The inset is too small. A larger spectral excerpt needs to be shown – not just in the SI. The way the single peak is shown without the rest of the spectrum is suboptimal – and extremely unusual in the field. A larger spectral section allows to assess spectral resolution, S/N, line splittings etc. much better.”:

Author reply: We now show a strip from the spectrum in Figure 4, to better show the spectral resolution.

Minor points:

Reviewer "- Experimental section: Please provide details about decoupling in ssNMR experiments, etc.”:

Author reply: Details about the decoupling in ssNMR experiments (83 kHz SPINAL decoupling) have been included in the text.

Reviewer "- „based on a reinforcement value measured during the simulation (See ref. ?? for a recent perspective on EVO-MD)“ -> Please check carefully all refs also: How can there be a recent perspective on Evo-MD, which is described for the first time in this manuscript??"

Author reply: Reference fixed. The perspective quoted covers a broad topic — physics-based inverse design of biopolymers interacting with complex liquid phases — discussing existing structure-focused protein design methods such as Rossetta, and refers to the pre-existing archive version of this manuscript as an emerging alternative approach on how to additionally handle sequence design in cases where protein functionality is predominantly determined by optimizing interactions with complex fluids and fluid interfaces (e.g., protein condensates and lipid membranes). The perspective also covers physics-based optimization of simulations system setup (e.g., optimizing lipid membrane or solvent composition) and biomolecular force fields, as well as integration with experimental data. We have renamed the 'perspective on EVO-MD' to 'perspective on physics-based optimization' to better reflect its content."

Reviewer "-Fig. S11. The PDSM mixing time was 15 seconds. ??? Normally I would expect a mixing time in the ms range."

Author reply: The PDSM mixing time was chosen in order to affect transfer over 6.9 angstrom. This is now properly explained in the text.

Review: *Physics-Based Evolution of Cholesterol-Attracting Transmembrane Helices: Deciphering Cholesterol Attraction in Native Membrane Proteins*

Summary

In this paper, Methorst et al. investigate the physical principles underlying cholesterol attraction to transmembrane helices using a combination of coarse-grained (CG) and all-atom (AA) molecular dynamics (MD) simulations. Some selected results are further confirmed with experiments. The coarse-grained simulations are driven by an evolutionary algorithm that optimizes the sequence of a single transmembrane helix to increase the cholesterol content found around it. This simulation strategy allows the authors to survey the large sequence space and sequences that optimally attract cholesterol.

The EvoMD results reveal that cholesterol attraction is largely governed by two physical principles: hydrophobic slenderness and hydrophobic mismatch. Hydrophobic slenderness refers to helix sequences, which largely consist of hydrophobic amino acids with small side chains such as valine or isoleucine. According to the findings presented here a larger hydrophobic mismatch promotes cholesterol binding through a secondary effect termed 'enforced snorkeling'. To compensate for the hydrophobic mismatch the cholesterol molecules tilt towards the transmembrane helix and bind the hydrophilic residues contributing to the mismatch. This tilting is similar to snorkeling – the spontaneous tiling of cholesterol/lipids - and hence termed enforced snorkeling spontaneously. Since the energy penalty for snorkeling in cholesterol is much less compared to PC lipids cholesterol molecules preferentially collect around the helix. This binding mode is confirmed by solid-state NMR. We note that these principles directly contradict the notation that CRAC/CARC motifs are indicative of cholesterol attraction as often postulated in literature.

Utilizing these physical insights the authors design two realistic sequences for optimal cholesterol attraction. As discussed by the authors the EvoMD algorithm converges to fairly unrealistic sequences that feature non-helical amino acids such as proline, high net charge, and too-short hydrophobic block length. They go on to measure the free energy of bonding between these helices and a single cholesterol molecule. The same simulation is repeated for γ M4 peptide, which features a CRAC motif commonly used to identify cholesterol attraction. Interestingly, the authors find that the new sequences bind cholesterol stronger when compared to the peptide featuring the CRAC motif. Perhaps even more surprising even the peptide mutant, which does not feature the CRAC motif binds cholesterol strongly.

This finding leads the authors to further investigate how predictive the CRAC motif is in predicting cholesterol attraction. To this end, they trained a Convolutional Neural Network (CNN) on the simulated data to predict the fitness score for arbitrary protein sequences. Subsequently, fitness scores were predicted for the entire TmAlphaFold database. They noticed that there is no correlation between the CRAC motif and cholesterol attraction as quantified by the fitness score. Furthermore, there even seems to be a weak anticorrelation. Ultimately, when analyzing the database it is found that the fitness does not correlate with the TMD length indicating that the hydrophobic mismatch plays an insignificant role in biological systems.

Strengths

In our view, the paper presents an interesting new take on Cholesterol-Protein interaction. Beyond that, the utilization of a GA in a high-throughput MD approach serves as an interesting example of the successes and limitations of such methods in theoretical biophysics. As such we consider the publication appropriate for Nature Communications.

Furthermore, the authors have confirmed their hypothesis that ‘enforced snorkeling’ due to hydrophobic mismatch leads to higher cholesterol attraction to a reasonable degree and also succeeded in calling into question the applicability of CRAC/CARC motifs to identify cholesterol attraction.

Weaknesses

As detailed below we consider the link between cholesterol attraction and amino acid size (i.e. hydrophobic slenderness) insufficiently proven and not separated clearly from the hydrophobic mismatch. The following questions remain open: What is the relative strength of each driving factor? Is the hydrophobic slenderness just a passive packing effect? Separating the two contributions seems particularly relevant when considering that hydrophobic mismatch plays a minor role when considering biologically plausible helices.

Furthermore, the authors strongly emphasize the success of EvoMD in identifying optimal cholesterol attractors. Based on these results it is suggested that EvoMD might be generally applicable to this kind of problem. However, considering that the EvoMD protocol fails to predict realistic sequences, we consider this view overly optimistic. As the authors state the limitations arise from the use of a coarse-grained model. However, it would have been more convincing if the fitness score had been adopted to include for example a penalty for placing a PRO in the helix. The same for charge balancing, and block length. In its current state EvoMD requires some expert knowledge and significant interpretation in order to derive any useful sequences from it.

Detailed Comments

1. The system sizes for the coarse-grained simulations are surprisingly small and significantly smaller than the usual membrane sizes considered even for parametrization purposes. Therefore, we consider it important to verify how much the fitness score changes with system size. Investigating how the fitness score changes with progressively larger systems for at least several protein sequences generated by the GA seems easy enough of a test.
2. In our view, the link between Cholesterol attraction and the size of the amino acid is insufficiently established. It appears to be true that there is a correlation between fitness and some smaller amino acids (e.g. PRO, VAL). However, the bond scaling analysis seems unnecessarily artificial. It also seems suspicious that the smallest hydrophobic amino acid Alanine is not present to a significant extent. Instead, it would make more sense to correlate the fitness with the SASA or SASA inferred volume of the hydrophobic domain. Ideally, this analysis could be done on the all-atom level after backmapping. If it is truly just the slenderness there should be a pretty good correlation with the SASA.

3. Can the authors comment on why V, and P, occur at positions 1, and 2 of the helix sequence according to Figure 3A? This part of the peptide should be in the hydrophilic / head-group region of the membrane. Therefore, it is strange to see such a significant population in that position.
4. From the analysis of predicted fitness scores for the TmAlphaFold database, the authors conclude that hydrophobic mismatch plays a minor role in real biological systems and instead cholesterol attraction must be driven by hydrophobic slenderness. However, no actual proof is offered at this stage. We suggest correlating the fitness score with the occurrence of smaller hydrophobic amino acids and/or the SASA of the hydrophobic domain.
5. One key piece of evidence is the all-atom PMF presented in Figure 3. There is a growing consensus in the MD community that PMFs generated by umbrella sampling (US) are very susceptible to how initial starting configurations are generated, the window spacing, and the force constant. A recent study by Aho *et al.* suggests even for well-defined 1D reaction coordinates there can be as much as 2-20 kcal/mol in difference. In light of these results, it is surprising that the presented PMFs appear to be only one replicate. Usually, a single replicate is insufficient and not in line with the best practices of the field. We suggest adding at least two replicates and providing the histograms in the SI so that the sufficient overlap criterion can be verified.
6. The description of the training procedure for the NN is incomplete. The following question should be answered in the SI or method section: What exactly were the CNN inputs? How big were the training and testing data sets? How does CNN perform on both?
7. Usage of term 'sensing': Sensing requires recognition and response. As we do not see evidence of any response, we suggest avoiding this term.
8. Some of the figures and captions need to be improved:
 - a. Fig. S3: The gray background of panels D,F,E,G is confusing. Does it have any meaning?
 - b. Fig. S3: broken caption: "non-CARC (F- ζ A)"
 - c. Fig. S8: What are the colors? What does this plot show (averaged over time or a single snapshot?) How is the % calculated?
 - d. Fig. S15: What are the black below molecules?
 - e. Fig. S16-S19: The contrast of the especially yellow lines is very poor and therefore hardly visible.
 - f. p16/17: References broken to Fig. 5
9. From the current manuscript, it is not obvious if and where the code and data are available. We suggest providing the evoMD code as well as the NN code online. In addition, providing the results of the evoMD at least in terms of sequences and fitness score seems appropriate.